# SCALING LAWS FOR PARAMETER PRUNING IN LLMS

## ABSTRACT

Scaling up model parameters and training data consistently improves the performance of large language models (LLMs), but at the cost of rapidly growing memory and compute requirements, which makes deployment on resource-limited hardware infeasible. *Model pruning*, a widely used compression technique, reduces inference costs by removing redundant parameters. However, its impact on downstream performance remains unpredictable and is typically assessed only through costly empirical sweeps. To address this gap, we introduce *pruning laws* – simple and interpretable scaling relations that connect a pruned LLM's post-pruning performance to its unpruned performance and pruning ratio. Across five LLMs (2.7B–13B parameters), three pruning strategies (unstructured, width, and depth), and eight diverse tasks, we show that pruning laws achieve strong predictive accuracy (average extrapolation error $< 7\%$), reliably quantify performance degradation, and identify critical pruning thresholds beyond which recovery is infeasible. Moreover, we demonstrate that the same laws transfer universally across architectures, pruning methods, and even unseen models in zero-shot and one-shot setups. These results provide both researchers and practitioners with a principled framework to select pruning strategies, estimate safe pruning ratios without exhaustive tuning, and deploy LLMs efficiently under real-world compute and latency constraints.

## 1 INTRODUCTION

In recent years, there has been growing interest in understanding how the size of pre-training models and datasets impacts the downstream performance of large language models (LLMs). *Neural scaling laws* (Kaplan et al., 2020; Hoffmann et al., 2022; Muennighoff et al., 2023) formalize the relationships between model performance, size, data, and compute, showing that performance improves predictably as these factors scale. However, recent studies (Faiz et al., 2024; Diaz & Madaio, 2024; Villalobos et al., 2024) also show that scaling leads to nearly linear growth in computational costs, highlighting the need for more efficient LLMs that retain accuracy under limited resources.

*Model pruning* has emerged as a widely used approach to compress LLMs into smaller and more efficient counterparts. Both unstructured pruning (Frantar & Alistarh, 2023) and structured pruning methods such as depth pruning (Yang et al., 2024), width pruning (Ashkboos et al., 2024), and calibration-free pruning (Sengupta et al., 2025) have attracted significant attention. These methods remove redundant components from pre-trained LLMs, often with minimal loss in performance. Yet, despite their adoption, there has been no systematic framework for understanding how pruning impacts downstream performance across models, tasks, and strategies. To address this gap, we introduce **scaling laws for parameter pruning in LLMs** (henceforth *pruning laws*), providing an analytical framework for evaluating the scalability and effectiveness of pruning.

**RQ1.** How does pruning affect downstream performance across task categories?
**RQ2.** How do pruning strategies trade off between efficiency and accuracy?
**RQ3.** Can pruning laws accurately predict performance, even for unseen models and methods?
**RQ4.** How can pruning laws guide principled pruning decisions for practitioners?

**Model and method coverage.** Our analysis spans five off-the-shelf LLMs – OPT-2.7B, OPT-6.7B, OPT-13B, LLaMA-7B, and LLaMA-13B, widely used as pruning baselines. These cover

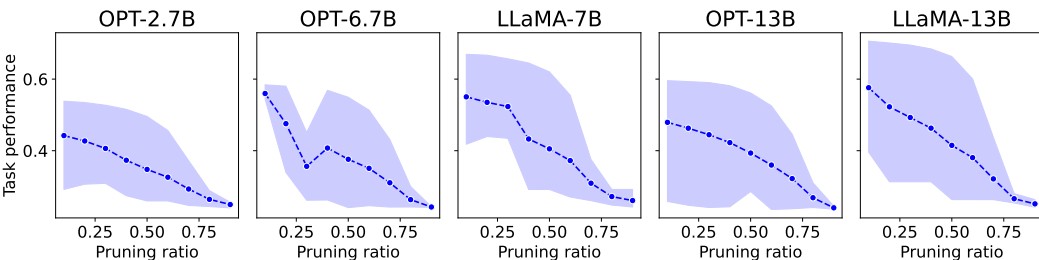

Figure 1: Average downstream performance of different LLMs at varying pruning ratios. The shaded region shows the variance across pruning strategies, emphasizing the need for systematic evaluation of pruning effects. Detailed results are available in Tables 4 and 5 (Appendix D).

two model families and multiple parameter scales, enabling systematic study of pruning effects. To test generalization, we further apply pruning laws to newer architectures including LLaMA-3.1 (Dubey et al., 2024) and Phi-3 (Abdin et al., 2024), and to pruning methods not seen during training (e.g., SlimGPT (Ling et al., 2024), SVD-LLM (Wang et al., 2024)). Each model is pruned using unstructured, width, and depth pruning across retention ratios (defined as '1 - pruning ratio') from 10%–90%. Evaluation covers eight downstream tasks: five commonsense reasoning, two language modeling, and one QA. Unlike prior work, we do not apply recovery fine-tuning; instead, we directly fit pruned model performance $\mathcal{L}$ as a function of base accuracy $\mathcal{L}_0$ and retention ratio $(1 - r)$ via a power law (Section 3). Below we highlight the key empirical observations:

**RQ1. Impact on downstream performance.** Across all eight tasks (Figure 1), models retain at least 80% of baseline accuracy with up to 50% pruning, beyond which, performance decays following a power law. At a task level, reasoning tasks are highly resilient, with performance decaying slowly, retaining strong accuracy even at 50% pruning. QA tasks are most fragile, showing sharp drops with only ∼70% accuracy preserved at moderate pruning. Language modeling lies in between, degrading more gradually.

**RQ2. Computational benefits.** Depth pruning yields the largest inference gains, up to 5× speedup at 90% pruning for OPT-13B, but suffers poor fit and high test error (∼0.1). Unstructured pruning preserves accuracy best (test error ≈ 0.02–0.06) but offers only ∼1.2× speedups without specialized hardware. Width pruning provides a balanced trade-off, achieving moderate accuracy retention and speedups of ∼1.3–1.4×. (c.f. Figure 2).

**RQ3. Generalization to newer models and pruning methods.** Across five base LLMs (2.7B–13B) and multiple pruning strategies, pruning laws achieve average extrapolation errors < 7%. Zero-shot transfer to unseen architectures (e.g., LLaMA-3.1, Phi-3) yields errors of 0.04–0.08, while one-shot calibration refines them further (e.g., SlimGPT error on LLaMA-1-7B drops from 0.13 → 0.05). This universality highlights that pruning laws are simple, portable, and robust for real-world planning.

**RQ4. Practical implications of pruning laws.** Pruning laws enable forward planning of pruning ratios and strategies. For instance, at 50% pruning, OPT-2.7B loses ∼15% accuracy versus ∼18% for OPT-6.7B. Reasoning tasks tolerate aggressive pruning (up to 60–70%) before collapse, whereas QA and language modeling require conservative ratios (< 40%). Practitioners can rely on zero-shot reuse of coefficients for immediate predictions, or one-shot calibration when limited evaluation budget exists, avoiding costly trial-and-error.

The pruning laws introduced here offer the first systematic framework for understanding how pruning impacts LLMs across models, tasks, and methods. They provide interpretable and practical guidelines for compressing LLMs under real-world compute and latency constraints[1]. We have reported a list of **FAQs** regarding pruning laws and their responses in Appendix A.

## 2 RELATED WORK

**Model pruning for parameter efficiency.** Despite the impressive capabilities of LLMs such as LLaMA (Dubey et al., 2024) and DeepSeek (DeepSeek-AI et al., 2024) across diverse tasks, including natural language inference, complex reasoning, summarization, translation, and code generation, their large-scale deployment remains hindered by substantial computational resource demands. Model

---

[1]Source code, datasets, and pruned model checkpoints will be released publicly upon acceptance. Supplementary material contains the datasets used for fitting pruning laws.

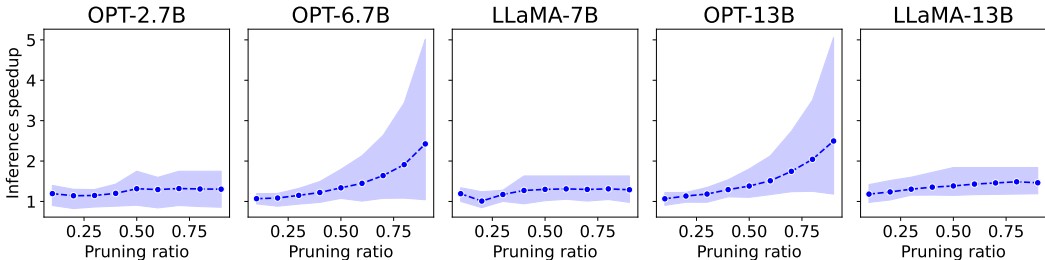

Figure 2: Inference speedup of different models at varying pruning ratios on an autoregressive language modeling benchmark. Detailed results are available in Table 6 (Appendix D).

pruning is a common technique to reduce the parameter count in pre-trained models, improving their computational efficiency and speed. It generally falls into two main categories: unstructured and structured pruning. *Unstructured pruning* focuses on removing individual weights (Frantar & Alistarh, 2023; Sun et al., 2023) from pre-trained models. Despite their ability to retain performance post-pruning, unstructured pruning often demands hardware-specific optimizations and may not always lead to substantial computational benefits. Conversely, *structured pruning* eliminates entire channels or components, making it more suitable for a broader range of hardware configurations. Contemporary structure pruning methods like SliceGPT (Ashkboos et al., 2024), layer collapse (Yang et al., 2024) use a small calibration dataset to assess the importance of different components of a pre-trained model and removes them subsequently, if found unimportant. Sengupta et al. (2025) proposed a policy-driven calibration-free model pruning method and argued that LLMs can withstand even when pruned by a random subset of the pre-trained components.

**Neural scaling laws for LLMs and parameter-efficient models.** The study of scaling laws dates back to Cortes et al. (1993), who analyzed generalization error as a function of training steps and dataset size. Subsequent studies such as Hestness et al. (2017) and Rosenfeld et al. (2019) examined the scaling of deep neural networks across models and data regimes. Kaplan et al. (2020) proposed the functional form $L \sim N^{-\alpha} + D^{-\beta}$ (*Kaplan scaling law*), linking test loss to parameter count $N$ and dataset size $D$, while Hoffmann et al. (2022) introduced *Chinchilla scaling laws*, emphasizing compute-optimal trade-offs $C \sim N \cdot D$ and showing that smaller models trained on more tokens can outperform larger ones. Later, Caballero et al. (2023) argued that Kaplan's formulation, being monotonic, fails to capture emergent behaviors and phase transitions in pre-trained Transformers (Vaswani, 2017).

Recent works extend scaling laws to parameter-efficient settings. Busbridge et al. (2025) proposed *distillation scaling laws*, modeling student performance distilled from larger teachers under varying compute budgets, and highlighting when distillation is preferable to supervised pretraining. Kumar et al. (2024) introduced *precision-aware scaling laws*, showing that training in reduced precision lowers effective parameter counts and improves scaling efficiency. Chen et al. (2024) further analyzed recovery fine-tuning for LLMs pruned with structured methods, focusing on the extent of recovery needed to mitigate pruning-induced degradation. While these works capture important trends, they do not yet provide a unified framework for understanding scaling under pruning, which is the focus of our study.

**Pruning law in broader literature.** To contextualize pruning within the broader family of model compression techniques, we compare our proposed pruning laws with two recently introduced frameworks: *distillation scaling laws* (Busbridge et al., 2025) and *quantization scaling laws* (Kumar et al., 2024). Both approaches attempt to predict post-compression performance using analytical functions of compute, precision, or student-teacher capacity gaps. However, our empirical results demonstrate that pruning scaling laws not only yield tighter empirical fits but also exhibit greater generalizability across model families, tasks, and compression regimes. Distillation scaling laws model performance as a function of compute allocation between teacher and student models. While informative, their predictive utility is limited by sensitivity to student-teacher pairings and reliance on retraining, often from scratch. Furthermore, the effectiveness of distillation scaling is contingent on access to a high-quality teacher, which may not always be feasible in deployment-constrained settings. Quantization scaling laws (Cao et al., 2024; Kumar et al., 2024), on the other hand, typically assume fixed architectures and specific quantization backends. While these approaches offer valuable characterizations, they are often tightly coupled to hardware and require extensive calibration.

In contrast, our pruning laws offer an architecture-agnostic, task-aware framework that models performance degradation as a function of pruning ratio. They are directly applicable to both structured and unstructured pruning, across varying model sizes and families. Pruning laws further support the definition of *critical pruning thresholds*, providing actionable guidance for practitioners on how much pruning can be applied before recovery becomes infeasible. These properties make pruning laws not only more interpretable but also more readily deployable in real-world scenarios where compression needs to be adaptive, fast, and data-efficient.

## 3 METHODOLOGY

### 3.1 PARAMETRIZATION OF THE LLM PRUNING LAW

With pruning laws, we propose a series of analytical methods for estimating the performance of LLM post-pruning on a variety of downstream tasks. For all downstream tasks, we assume that the performance of a model on a task is captured by a metric, which is bounded (e.g., model accuracy); in other words, higher the performance of a model, the better it is. Building on prior studies (Kaplan et al., 2020; Hoffmann et al., 2022) that establish scaling laws for LLM pre-training, we formulate a relationship between the performance of a pruned model and its corresponding base model through a law defined by two key parameters: the performance of the base model on a task (denoted by $\mathcal{L}_0$) and the pruning ratio used to prune the model (denoted by $r \in (0, 1)$); we denote the relationship by $\mathcal{L} := \mathcal{L}(\mathcal{L}_0, r)$, where $\mathcal{L}$ represents the performance of the pruned model. Our proposed pruning law can be used to determine the optimal value of $r$ needed to obtain a well-performing pruned model, maximizing the performance retainment post-pruning.

The functional form of our parametrization is described by the following equation:

$$\mathcal{L}(\mathcal{L}_0, r) = \mathcal{L}_0 P_0 (1-r)^{\alpha} \tag{1}$$

where $\alpha$ and $P_0$ are real numbers. The exponent $\alpha$ controls how quickly performance decays as the pruning ratio $r$ increases. The coefficient $P_0$ is a *bias term* that encodes the boundary conditions of the law: although $(1 - r)$ tends to 1 as $r$ approaches 0, the performance of the pruned model at extremely small pruning ratios may not exactly match the base model's performance. Factors such as evaluation noise, subtle architecture-dependent effects and differences among pruning methods can introduce a systematic offset at $r \to 0$. Consequently, $P_0$ captures these offsets and depends on the particular model, pruning method and task being considered.

**Levels of fit.** In our analysis, we fit the pruning law at several levels of aggregation:

1. **Task level:** we fit a single $(\alpha, P_0)$ pair for each task, pooling data across all models and pruning methods. In this case, $P_0$ reflects the average boundary-condition bias for that task.

2. **Method–task level:** we fit the law separately for each pruning method and task across all models. Here $P_0$ absorbs both task-specific and method-specific biases (e.g., unstructured vs. structured pruning may yield different offsets near $r = 0$).

3. **Model–task level:** we fit the law for each model and task, pooling across methods. $P_0$ then captures architecture-specific bias for that task, indicating how resilient a given model is on that task when lightly pruned.

4. **Method–model–task level:** we fit the law for each combination of model, pruning method and task. This most granular fit yields a distinct $P_0$ for every configuration, which serves as the local intercept for that specific pruning setup.

This choice of functional form is motivated by the following *feasibility conditions* for a pruning law:

- We formulate the pruning law as a power law with respect to the retention ratio. Unlike some pre-training scaling laws, we require the law to be *scale invariant* with respect to $(1 - r)$, i.e., $\mathcal{L}(\mathcal{L}_0, r)$ must be a homogeneous function of $(1 - r)$. Such a form allows us to derive optimal decision regions for choosing $r$ given a target performance drop.
- The post-pruning performance should decrease as the pruning ratio $r$ increases, i.e., $\frac{\partial \mathcal{L}}{\partial r} < 0$. In Equation 1, this behaviour is captured by requiring $\alpha > 0$.
- The functional form should capture the *iterative nature* of many pruning methods: pruning a model by $r_1$ followed by further pruning by $r_2$ should yield a performance proportional to a single pruning with overall pruning ratio $1 - (1 - r_1)(1 - r_2)$ (Sengupta et al., 2025; Yang et al., 2024).

Through extensive experiments (see Section 5), we find that the proposed form in Equation 1 satisfies these feasibility conditions.[2]

## 3.2 FITTING THE PRUNING LAW: ORDINARY LEAST SQUARES FOR LINEAR REGRESSION

Taking logarithms on both sides of Equation 1 gives:

$$\log \mathcal{L} = \log \mathcal{L}_0 + \alpha \log(1-r) + \log P_0, \tag{2}$$

Therefore, fitting the pruning law reduces to a linear regression in log space. In this formulation, $\alpha$ is the slope (multiplying $\log(1-r)$) and $\log P_0$ is the intercept, capturing the boundary-condition bias discussed above. The regression is performed on the transformed variables $r' := 1 - r$ with $\log P_0$ as the intercept. To learn $\alpha$ and $P_0$, we use the standard ordinary least squares (OLS) method (Zdaniuk, 2014), assuming additive Gaussian noise. The regression model can thus be stated as:

$$\log \mathcal{L} = \log \mathcal{L}_0 + \alpha \log(1-r) + \log P_0 + \epsilon_{\text{noise}}, \tag{3}$$

where $\epsilon_{\text{noise}} \sim \mathcal{N}(0,1)$.[3]

## 4 EXPERIMENTAL SETUP

To develop the proposed pruning laws, we compress and evaluate a range of LLMs, including OPT (Zhang et al., 2022) at 2.7B, 6.7B, and 13B parameter scales, and LLaMA-2 (Touvron et al., 2023) at 7B and 13B. Moreover, for evaluating the universality of the pruning laws on more recent architectures, we consider LLaMA-3.1-8B (Dubey et al., 2024) and Phi-3-mini-4K-Instruct (Abdin et al., 2024) models. All pre-trained model checkpoints are obtained from Hugging Face [4]. We experiment with three pruning methods: (1) **SparseGPT** (Frantar & Alistarh, 2023) for unstructured weight pruning, (2) **LaCo** (Yang et al., 2024) for structured depth pruning (layer collapse), and (3) **SliceGPT** (Ashkboos et al., 2024) for structured width pruning. Additional testing is conducted with two further depth pruning methods – **SlimGPT** (Ling et al., 2024) and **LLM Pruner** (Ma et al., 2023) and two width pruning methods including **SVD-LLM** (Wang et al., 2024) and **PruneNet** (Sengupta et al., 2025). For each method, we apply a comprehensive range of sparsity levels with pruning ratios of $\{10\%, 20\%, \ldots, 90\%\}$. We evaluate the pruned models on eight downstream tasks using the LM Evaluation Harness (Gao et al., 2024) [5], performing all evaluations in a zero-shot setting.[6] The tasks span three categories:

- **Reasoning**: PIQA (Bisk et al., 2020), WinoGrande (Sakaguchi et al., 2021), HellaSwag (Zellers et al., 2019), ARC-e and ARC-c (Clark et al., 2018).
- **Question-answering**: CoQA (Reddy et al., 2019)
- **Language modeling**: WikiText (Merity et al., 2016) and LAMBADA (Paperno et al., 2016).

For classification-style tasks, including reasoning and QA, we report *accuracy*. For language modeling tasks, we compute *perplexity*, which is unbounded $(1, \infty)$; thus, we report the inverse log-perplexity (i.e., $1/\log(\text{ppl})$) to map the metric to a $(0,1)$ range for consistency with other evaluations. In addition to these individual tasks, we calculate the average task performance, highlighted by **Average**.

**Evaluating pruning laws.** For evaluating the fitted pruning laws on in-distribution data points (e.g., same model/pruning method used in training and testing), we use *rolling-style* test data, where for each pruning ratio $r \in [20\%, 90\%]$, we train the pruning laws on pruning ratios $\{10\%, \cdots, r\%\}$ and test the pruning laws on $\{r\% + 10\%, \cdots 90\%\}$. Finally, we calculate the average root mean square error (RMSE) between the predicted and ground truth over all the test datasets. For evaluating the extrapolation capability of the pruning laws in out-of-distribution setup, we adopt two extrapolation evaluation strategies – *zero-shot extrapolation*, where we use the fitted parametric functions to test on unseen data, and *one-shot extrapolation*, where we use the $\alpha$ coefficient from the fitted parametric

---

[2]Refer to Section B.1 of the Appendix for a proof that equation 1 satisfies the iterative pruning condition.

[3]A similar implementation with robust estimation Huber loss (Huber, 1992) is also performed, but due to very marginal differences omitted from this paper.

[4]https://huggingface.co/models

[5]Task descriptions are provided in Appendix C.

[6]We categorically ingore knowledge-extraction tasks like MMLU (Hendrycks et al., 2020) in our study, highlighting the reason in a detailed section in Appendix E.

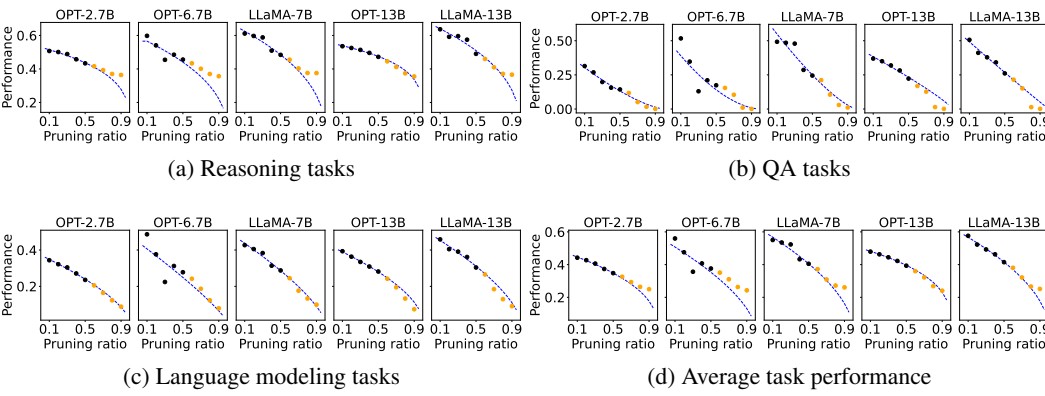

(a) Reasoning tasks

(b) QA tasks

(c) Language modeling tasks

(d) Average task performance

Figure 3: Fitted pruning laws for downstream performance of pruned LLMs. **Black** and orange points highlight the training and testing data points, respectively, with blue line indicating the scores predicted by our fitted pruning laws.

function, but the bias term $P_0$ is re-estimated from pruning the model on a single pruning ratio. Precisely, let $\alpha$ and $P_0$ denote the fitted coefficients. For a given pruning ratio $r$, we prune the LLM, obtain its performance $\mathcal{L}_r$, and re-estimate the bias term $\hat{P}_0 = \frac{\mathcal{L}_r}{\mathcal{L}_0 \times (1-r)^\alpha}$. While zero-shot extrapolation setup is intended to verify the universality of the pruning laws, one-shot extrapolation is particularly useful for verifying the flexibility of pruning laws for testing out-of-distribution models or pruning strategies, when a single point enumeration is practically possible.

# 5 EXPERIMENTAL RESULTS

## 5.1 MAIN RESULTS

**Impact of pruning on model performance.** Pruning has a clear and systematic effect on the performance of LLMs, with its impact varying across model sizes, architectures, and task categories. As highlighted in Tables 4 and 5 of Appendix D, larger models such as LLaMA-13B and OPT-13B exhibit higher resilience at lower pruning ratios, maintaining competitive accuracy up to 30-40% pruning, whereas smaller models like OPT-2.7B and OPT-6.7B degrade much more rapidly, showing steep declines in QA and language tasks even at moderate pruning levels. Among task types, reasoning tends to be more robust to pruning than QA or language modeling, with models like LLaMA-7B and LLaMA-13B retaining relatively strong reasoning scores at 50% pruning, despite significant losses in QA and language modeling. However, when pruning ratios exceed 70-80%, performance consistently collapses across all models, with average scores approaching random baselines. These results highlight two consistent trends: (i) larger models offer greater tolerance to pruning due to redundancy in their parameterization (behaviors acknowledged in the prior literature like Frantar & Alistarh (2023)), and (ii) reasoning-oriented tasks degrade more gracefully under

| Task | $\alpha$ | $P_0$ | Adj $R^2$ | F Statistic | Test Error |
|------|----------|-------|-----------|-------------|------------|
| **QA** | $2.42 \pm 0.13$ | $1.45 \pm 0.14$ | 0.89 | 348.48 | 0.04 |
| **Reasoning** | $0.22 \pm 0.02$ | $0.95 \pm 0.02$ | 0.79 | 170.83 | 0.06 |
| **Language** | $0.73 \pm 0.02$ | $0.76 \pm 0.03$ | 0.96 | 947.9 | 0.03 |
| **Average** | $0.35 \pm 0.02$ | $0.85 \pm 0.02$ | 0.88 | 316.37 | 0.05 |

(a) Task-level

| Task | Model | $\alpha$ | $P_0$ | Adj $R^2$ | F Statistic | Test Error |
|------|-------|----------|-------|-----------|-------------|------------|
| | OPT-2.7B | $0.28 \pm 0.03$ | $0.85 \pm 0.03$ | 0.93 | 109.53 | 0.03 |
| | OPT-6.7B | $0.34 \pm 0.05$ | $0.87 \pm 0.06$ | 0.83 | 39.88 | 0.10 |
| **Average** | LLaMA-7B | $0.38 \pm 0.04$ | $0.86 \pm 0.05$ | 0.90 | 74.55 | 0.07 |
| | OPT-13B | $0.34 \pm 0.02$ | $0.87 \pm 0.02$ | 0.98 | 375.15 | 0.02 |
| | LLaMA-13B | $0.4 \pm 0.04$ | $0.83 \pm 0.04$ | 0.94 | 126.08 | 0.05 |

(b) Model-task level

| Task | Method | $\alpha$ | $P_0$ | Adj $R^2$ | F Statistic | Test Error |
|------|--------|----------|-------|-----------|-------------|------------|
| | Unstructured | $0.47 \pm 0.03$ | $1.22 \pm 0.03$ | 0.90 | 370.20 | 0.15 |
| **Average** | DepthPruning | $0.13 \pm 0.04$ | $0.54 \pm 0.04$ | 0.20 | 12.21 | 0.10 |
| | WidthPruning | $0.39 \pm 0.04$ | $0.83 \pm 0.04$ | 0.70 | 103.11 | 0.10 |

(c) Method-task level

Table 1: Coefficients of the task-level (a), model-task level (b) and method-task-level (c) pruning laws of the form $\mathcal{L} = \mathcal{L}_0 P_0 (1 - r)^\alpha$. We report the coefficients estimated with ordinary least square (OLS) fits, along with the standard errors. To evaluate the goodness-of-fit, we calculate adjusted $R^2$, F statistic score. We calculate the test error as the average root mean square error on the test data. All task coefficients are reported in Table 9 of Appendix D.

compression compared to language-modeling tasks, suggesting that pruning disproportionately affects linguistic knowledge over logical problem-solving abilities.

**Pruning laws for downstream performance.** Table 1 demonstrates pruning laws that characterize how downstream performance scales with pruning ratios. Across tasks (cf. Table 1a), the exponent $\alpha$, which controls the sensitivity to pruning, tends to be highest for QA ($\alpha \approx 2.4\text{-}2.7$) and smallest for reasoning ($\alpha \approx 0.2$). This indicates that the QA accuracy decays sharply with pruning, whereas reasoning remains relatively stable even under high compression. Language tasks fall in between ($\alpha \approx 0.7\text{-}0.9$), showing gradual degradation. The bias term $P_0$, which modulates retained performance, is typically close to one for reasoning tasks ($P_0 \approx 0.9\text{-}0.95$), reflecting their robustness. In contrast, it is substantially larger for QA ($P_0 \approx 1.4\text{-}1.7$), indicating that pruning induces disproportionately large performance drops. Importantly, these fits achieve high adjusted $R^2$ scores (0.88-0.99) and significant $F$-statistics, indicating that the pruning law explains a large fraction of the variance in performance with minimal error. The standard errors on the coefficients are small (typically $\pm 0.02$ -0.1), reinforcing robustness in parameter estimation, while the test errors remain low (0.02-0.06).

**Dependence of model architecture on effectiveness of pruning.** Across model families (c.f. Table 1b and detailed results reported in Table 7 of Appendix D), the pruning law maintains strong predictive accuracy, though the sensitivity to pruning varies with architecture and scale. Larger models such as LLaMA-13B and OPT-13B exhibit higher $\alpha$ values for average performance ($\alpha \approx 0.34\text{-}0.40$) compared to smaller OPT-2.7B ($\alpha \approx 0.28$), indicating smoother degradation due to greater redundancy. Bias terms remain stable around ($P_0 \approx 0.85 - 0.9$) across larger models, while smaller ones (OPT-6.7B, OPT-2.7B) show greater variability and higher test errors ($0.07 - 0.1$), implying reduced resilience. The high adjusted scores $R^2$ ($0.83 - 0.98$) and large $F$ statistics (ranging up to 375 for OPT-13B) confirm that the pruning law reliably captures architecture-specific behaviors (also validated in the fitted pruning laws in Figure 3), with deviations arising mainly in smaller models where compression stress is more acute. This suggests that larger models both empirically and analytically offer stronger buffers against pruning-induced degradation. The analytical behaviors confirmed by the pruning laws also affirm the empirical evidence found in the existing literature on model pruning.

**Does the pruning method matter?** Method-level coefficients reported in Table 1c (with detailed results in Table 8, Appendix D) reveal clear differences among pruning strategies – not only in their $\alpha$ and $P_0$ values, but also in overall goodness of fit. Unstructured pruning exhibits the strongest adherence to the pruning law, with consistently high adjusted $R^2$ ($\approx 0.9$), and large $F$-statistics (183-418). Depth pruning, by contrast, performs poorly, with very low adjusted $R^2$ (0.18-0.23), weak $F$-

| Model | Method | $\alpha$ | $P_0$ | Adj $R^2$ | F Statistic | Test Error |
|---|---|---|---|---|---|---|
| OPT-13B | | $0.01 \pm 0.0$ | $0.82 \pm 0.0$ | 0.94 | 118.19 | 0.01 |
| LLaMA-13B | | $0.02 \pm 0.0$ | $0.87 \pm 0.0$ | 0.97 | 294.22 | 0.00 |
| LLaMA-7B | Unstructured | $0.01 \pm 0.0$ | $0.81 \pm 0.0$ | 0.95 | 167.97 | 0.00 |
| OPT-6.7B | | $0.01 \pm 0.0$ | $0.84 \pm 0.0$ | 0.96 | 176.74 | 0.00 |
| OPT-2.7B | | $0.01 \pm 0.0$ | $0.78 \pm 0.0$ | 0.50 | 9.15 | 0.01 |
| OPT-13B | | $0.72 \pm 0.03$ | $0.94 \pm 0.03$ | 0.99 | 589.92 | 0.04 |
| LLaMA-13B | | $0.1 \pm 0.04$ | $0.64 \pm 0.04$ | 0.48 | 8.35 | 0.15 |
| LLaMA-7B | DepthPruning | $0.17 \pm 0.1$ | $0.81 \pm 0.11$ | 0.21 | 3.14 | 0.26 |
| OPT-6.7B | | $0.72 \pm 0.03$ | $0.96 \pm 0.03$ | 0.99 | 677.70 | 0.03 |
| OPT-2.7B | | $0.14 \pm 0.05$ | $0.73 \pm 0.05$ | 0.51 | 9.27 | 0.20 |
| OPT-13B | | $0.14 \pm 0.04$ | $1.02 \pm 0.05$ | 0.55 | 10.58 | 0.19 |
| LLaMA-13B | | $0.15 \pm 0.04$ | $0.95 \pm 0.04$ | 0.69 | 18.91 | 0.18 |
| LLaMA-7B | WidthPruning | $0.02 \pm 0.02$ | $1.02 \pm 0.02$ | -0.06 | 0.56 | 0.09 |
| OPT-6.7B | | $0.08 \pm 0.02$ | $1.08 \pm 0.03$ | 0.57 | 11.44 | 0.11 |
| OPT-2.7B | | $0.0 \pm 0.02$ | $1.15 \pm 0.02$ | -0.14 | 0.01 | 0.08 |

Table 2: Coefficients of pruning law for inference speedup of the form $\mathcal{L} = \mathcal{L}_0 P_0 (1 - r)^{\alpha}$.

statistics (10-14), and unstable bias terms ($P_0 \approx 0.25\text{-}0.7$), indicating that it introduces noise and fails to follow the law reliably. Width pruning falls in between: it yields moderate $\alpha$ values ($\approx 0.3\text{-}0.8$), relatively stable $P_0$ ($\approx 0.8\text{-}0.9$), and adjusted $R^2$ ranging from 0.62-0.86, with test errors around 0.1, suggesting reasonable but less consistent predictability compared to unstructured pruning. These results indicate that our pruning law not only generalizes well across tasks and models but also effectively discriminates between pruning strategies, providing quantitative justification for favoring unstructured or width pruning over depth pruning. Moreover, because depth pruning is highly architecture-dependent, its behavior remains more volatile across model families and pruning ratios, leading to especially erratic post-pruning performance.

**Influence of model pruning on inference speed.** Pruning substantially improves inference speedup, but the extent depends both on pruning ratio and pruning method. Depth pruning delivers the largest benefits, with speedups rising super-linearly at higher pruning ratios, e.g., OPT-13B and OPT-6.7B achieve over $5\times$ faster inference at 90% pruning. This effect is reflected in the scaling law coefficients, where depth pruning shows relatively high $\alpha$ values (e.g., $0.72 \pm 0.03$ for OPT-13B) that

amplify speedup with retention ratio. In contrast, unstructured pruning yields more modest but stable improvements, typically around $1.2\times$ across pruning levels, consistent with its very low $\alpha \approx 0.01$, indicating nearly flat scaling. Width pruning falls in between, with gradual increases up to $1.3$-$1.4\times$ at high pruning ratios. The goodness-of-fit metrics further validate these trends: unstructured pruning shows strong adjusted $R^2$ values ($0.94$-$0.97$) and near-zero test error (c.f. Figure 4), capturing its predictable linear speedup, while depth pruning exhibits both high variance in $P_0$ and weaker fits for smaller models, explaining the less consistent gains. Overall, pruning laws confirm that depth pruning can yield dramatic inference acceleration, though at the cost of stability, whereas unstructured and width pruning provide steadier, more predictable improvements.

## 5.2 UNIVERSALITY OF PRUNING LAWS

Our results indicate that the same functional form $\mathcal{L}_1 = \mathcal{L}_0 (1-r)^\alpha P_0$ transfers across *models*, *architectures*, and *pruning methods* with moderate testing error. Figure 5 clearly illustrates the consistently low in-distribution (ID) (when a model from same family is used in fitting the functions) test errors, as compared to out-of-distribution (OOD) (when none of the same-family models used in fitting) test errors. Particularly, for reasoning and language modeling tasks with larger models (13B sized), having same family models in the training data (in-distribution) reduces the test error by $50\%$.

To further assess the universality of our fitted pruning laws, we evaluate their reliability on completely out-of-distribution models and pruning methods. We prune LLaMA-3.1-8B and Phi-3-Mini-4K-Instruct models using unstructured and width pruning methods, respectively, with pruning ratios $\{10\%, \cdots, 90\%\}$ and calculate the test RMSE between the ground truth and performance predicted by pruning laws with $\alpha$ and $P_0$ coefficients obtained from Table 8 of

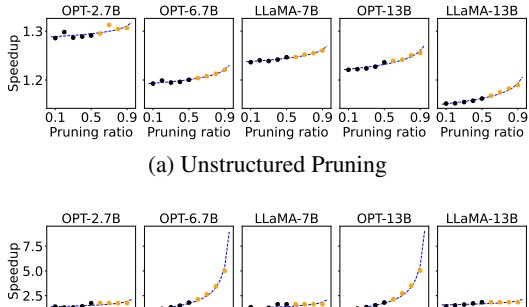

(a) Unstructured Pruning

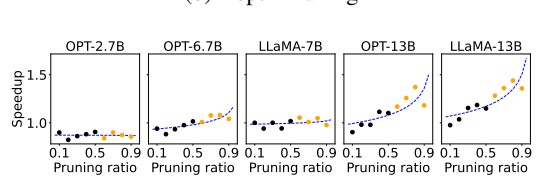

(b) Depth Pruning

(c) Width Pruning

Figure 4: Fitted pruning laws for inference speedup of pruned LLMs for different pruning methods.

Appendix D. In the one-shot extrapolation setup, we re-estimate the bias term $\hat{P}_0$ as described in Section 4 and predict the performance with coefficients $\alpha$ and $\hat{P}_0$. As highlighted in Table 3a, LLaMA-3.1-8B achieves an average zero-shot error of $0.04$, while Phi-3-Mini-4K-Instruct shows a slightly higher average error of $0.08$. Interestingly, the one-shot setup, where only the bias term $\hat{P}_0$ is recalibrated from a single data point, does not always reduce errors; instead, the average error increases to $0.06$ for LLaMA-3.1-8B and $0.10$ for Phi-3-Mini-4K-Instruct. This counterintuitive trend arises because $\alpha$ already captures the retention elasticity robustly across architectures, and introducing a single-point re-estimation of $P_0$ can inject noise rather than improve generalization, especially when the calibration point is not representative. These results reinforce that pruning laws are inherently universal: even without any recalibration, the fitted coefficients transfer effectively across unseen models and pruning methods, while one-shot calibration remains optional and may only be beneficial when calibration data are carefully chosen.

We further test universality by applying our fitted pruning laws to completely new pruning methods not used during training (Table 3b). We obtain the performances of LLaMA-1-7B model on reasoning tasks for depth pruning methods LLM Pruner (Ma et al., 2023) and SlimGPT (Ling et al., 2024) and two width pruning methods – SVD-LLM (Wang et al., 2024) and PruneNet (Sengupta et al., 2025), all numbers obtained from the corresponding papers. In the zero-shot setup, extrapolation errors remain small, ranging from $0.04$ for PruneNet to $0.13$ for SlimGPT and SVD-LLM. In the one-shot setup, where only $\hat{P}_0$ is recalibrated, errors decrease for most methods (e.g., LLM Pruner: $0.10 \rightarrow 0.06$, SlimGPT: $0.13 \rightarrow 0.05$, PruneNet: $0.04 \rightarrow 0.03$). Overall, these results reinforce the universality of pruning laws: they transfer across unseen pruning algorithms in zero-shot mode, while one-shot calibration may improve performance when the method aligns well with the retained elasticity.

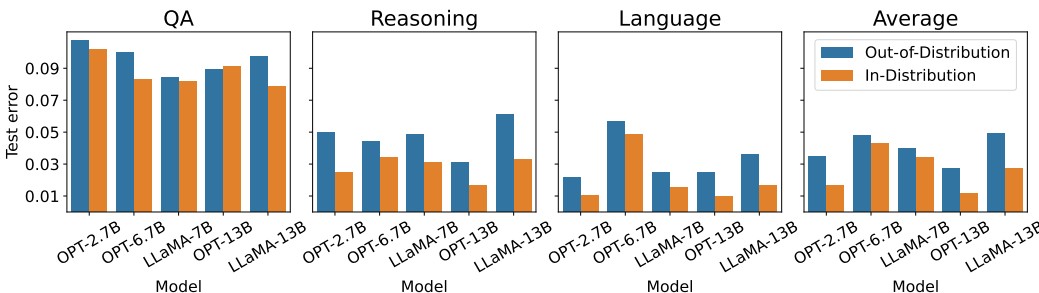

Figure 5: In-distribution (ID) and out-of-distribution (OOD) test error for different pruned LLMs. In the ID setup a model (e.g., LLaMA-7B) is tested while a different model from the same family (e.g., LLaMA-13B) is used in training. In the OOD setup, training and testing are performed on completely different model families.

| Model | Zero-shot error | | | | One-shot error | | | |
|---|---|---|---|---|---|---|---|---|
| | QA | Reasoning | Language | Average | QA | Reasoning | Language | Average |
| LLaMA-3.1-8B | 0.17 | 0.05 | 0.04 | 0.04 | 0.32 | 0.05 | 0.1 | 0.06 |
| Phi-3-Mini-4K-Instruct | 0.04 | 0.13 | 0.03 | 0.08 | 0.02 | 0.12 | 0.05 | 0.1 |

(a) Out-of-distribution Models

| Method Type | Pruning Method | Zero-shot error | One-shot error |
|---|---|---|---|
| DepthPruning | LLM Pruner | 0.1 | 0.06 |
| | SlimGPT | 0.13 | 0.05 |
| WidthPruning | SVD-LLM | 0.13 | 0.16 |
| | PruneNet | 0.04 | 0.03 |

(b) Out-of-distribution Methods

Table 3: Extrapolation errors with (a) different out-of-distribution LLMs and (b) pruning strategies.

Practically, this means practitioners can *predict* pruned performance for new model families and pruning strategies with zero data, and, if desired, achieve better estimation with a single-point calibration of $P_0$, supporting the claim that our pruning law is simple, portable, and universal.

### 5.3 PRACTICAL IMPLICATIONS OF PRUNING LAWS

Rather than relying on ad-hoc trial and error, our pruning laws provide a principled roadmap for model compression. Practitioners *seeking speed can turn to depth pruning*, those *prioritizing accuracy to unstructured pruning*, and those *balancing both to width pruning*. When *shifting to new models or pruning methods, zero-shot reuse of our coefficients enables immediate deployment*, while *one-shot calibration offers sharper estimates if a single evaluation is possible*. By aligning pruning choices with task sensitivity, *aggressive ratios for reasoning*, *conservative ones for QA and language modeling*, practitioners can achieve reliable efficiency gains without sacrificing performance. We provide more detailed practical guidelines in Appendix F for the readers to better synthesize our results for practical real-life deployment situations.

## 6 CONCLUSION

In this paper, we introduced pruning laws for LLMs that explore the impact of model pruning, offering new insights into the relationships between pruning ratios, performance metrics, and recovery fine-tuning. We provided practical guidelines for implementing model pruning in real-world applications, where both performance stability and scalability are crucial. This work lays the foundation for future research into adaptive, task-aware pruning methods, and the effects of pruning on long-context reasoning and generative capabilities. We also suggest investigating hybrid pruning strategies that combine structured and unstructured pruning to achieve a more balanced trade-off between computational savings and performance retention.

REPRODUCIBILITY STATEMENT

We have taken multiple steps to ensure the reproducibility of our findings. The experimental setup, including model selection and pruning strategies, is described in Section 4. All pre-trained model checkpoints (OPT-2.7B, OPT-6.7B, OPT-13B, LLaMA-7B, LLaMA-13B, and additional evaluation models) are publicly available through Hugging Face. Details of the pruning methods – SparseGPT (unstructured), LaCo (depth), and SliceGPT (width), and their implementation are provided in Section 3 and are already open-sourced in their respective code repositories. Comprehensive results, including fitted pruning coefficients, error statistics, and aggregation across models, methods, and tasks, are reported in Tables 4–6 and Appendix D. We evaluate using the LM Evaluation Harness with standardized metrics across eight downstream tasks, as detailed in Section 5. Upon acceptance, we will release source code, scripts for pruning and evaluation, configuration files, and pruned checkpoints, enabling full replication of both experiments and pruning law fitting (Section 3, Appendix D). Together, these resources ensure that our results can be reproduced, verified, and extended by the community.

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

## A   FREQUENTLY ASKED QUESTIONS (FAQS)

1. Why propose pruning laws when empirical benchmarks already exist?

   Empirical results alone do not generalize across models or pruning methods. Our pruning laws (Section 3) provide a compact analytical form with average extrapolation error $< 7\%$, enabling practitioners to predict outcomes without exhaustive sweeps.

2. Are the laws overfitting to specific models or datasets?

   No. The law is a simple two-parameter form ($\alpha$, $P_0$) fitted across five LLMs, three pruning strategies, and eight tasks. High adjusted $R^2$ values (0.83–0.98) confirm that it captures general scaling behavior rather than overfitting.

3. Why does depth pruning show weaker $R^2$ fits?

   Depth pruning produces flatter degradation curves with limited dynamic range, which reduces regression sensitivity. Nonetheless, predictive errors remain modest ($\leq 0.1$), and depth pruning delivers the strongest efficiency gains (up to $5\times$ speedup, Section D).

4. Can the pruning laws transfer to unseen models?

   Yes. On LLaMA-3.1 and Phi-3 models, zero-shot reuse of coefficients yields RMSE between 0.04–0.08. One-shot calibration further improves accuracy (e.g., SlimGPT error drops $0.13 \rightarrow 0.05$, Section 5).

5. How universal are the laws across pruning methods?

   Coefficients transfer across unstructured, width, and depth pruning. Even for unseen methods like SlimGPT and SVD-LLM, zero-shot predictions remain within 8% error, and one-shot calibration tightens the fit further.

6. Why exclude recovery fine-tuning in experiments?

   Our goal is to isolate the direct effect of pruning. Fine-tuning can mask pruning dynamics by recovering accuracy. Our laws are therefore conservative and can be extended in future work to model recovery separately.

7. Are the fits statistically significant?

   Yes. F-statistics are significant across most tasks ($p < 0.01$), and standard errors remain small ($< 0.05$ in many cases). Test errors are consistently low, supporting both robustness and reproducibility (Appendix D).

8. Are results specific to OPT and LLaMA models?

   While our main experiments use OPT and LLaMA, transfer tests with LLaMA-3.1 and Phi-3 confirm broader applicability. The functional form is architecture-agnostic.

9. How do tasks differ in pruning sensitivity?

   Reasoning tasks are robust ($\alpha < 0.3$, $P_0 \approx 0.95$), QA is brittle ($\alpha > 2$, $P_0 \approx 1.5$), and language modeling shows intermediate behavior ($\alpha \approx 0.7$).

10. Why is $\alpha$ important?

    $\alpha$ governs sensitivity to retention ratio. Large $\alpha$ values (QA) imply rapid collapse, while small $\alpha$ values (reasoning) imply gradual decay. This distinction explains task-level robustness.

11. What does $P_0$ represent?

    $P_0$ is a bias term reflecting baseline offset. High $P_0$ (e.g., QA tasks) indicates initial overestimation of performance before steep decline, while $P_0 \approx 1$ implies more linear decay.

12. Do smaller models prune differently than larger ones?

    Yes. Smaller OPT models lose accuracy faster (e.g., OPT-2.7B drops $\sim$15% at 50% pruning vs. $\sim$18% for OPT-6.7B). Larger models degrade more smoothly, making pruning more predictable.

13. How accurate are the predictions?

    Across models and tasks, extrapolation error is below 7%. For unstructured pruning, errors drop to 2–4%, demonstrating strong predictive fidelity.

14. Why does unstructured pruning preserve accuracy but not speedup?

    Sparse patterns require specialized kernels. Without them, FLOPs are not reduced efficiently, explaining why unstructured pruning shows negligible runtime gains despite low test error.

15. **How do width and depth pruning compare?**

    Width pruning balances accuracy and speed (moderate $\alpha$, $P_0$), while depth pruning maximizes speedup ($5\times$) but collapses accuracy faster at high pruning.

16. **Why not explore quantization or distillation alongside pruning?**

    Our focus is on pruning-specific dynamics. Distillation and quantization scaling laws exist separately (Busbridge et al., 2025; Kumar et al., 2024). Our pruning laws complement, rather than replace, these frameworks.

17. **Can pruning laws help practitioners in deployment?**

    Yes. Zero-shot reuse of coefficients gives immediate predictions for unseen models; one-shot calibration refines accuracy with a single evaluation. This reduces costly sweeps and guides safe pruning ratios (Appendix F).

18. **How reproducible are the results?**

    All experiments use public checkpoints, LM Evaluation Harness, and standard pruning methods. Detailed results, fitted coefficients, and scripts will be released upon acceptance (see Reproducibility Statement).

## B  THEORETICAL RESULTS

### B.1  FUNCTIONAL FORM LEADS TO PROPORTIONAL PERFORMANCE FOR ITERATIVE PRUNING

Here, we show that our proposed functional form in Equation 1 satisfies the last feasibility condition for iterative pruning methods. So, let $r_1 \in (0, 1)$ and $r_2 \in (0, 1)$ be two pruning ratios. Suppose a model is first pruned at ratio $r_1$, followed by a pruning by the same method at ratio $r_2$. It is easy to see that the overall pruning ratio (*i.e.*, a pruning ratio achieving the total pruning in one step) is $r_1 + r_2 - r_1 r_2$. Then, the performance obtained by the multi-step pruning is

$$
\begin{aligned}
\mathcal{L}_{\text{mutlistep}} &= \mathcal{L}(\mathcal{L}(\mathcal{L}_0, r_1), r_2) \\
&= \mathcal{L}(P_0 \mathcal{L}_0 (1 - r_1)^\alpha, r_2) \\
&= P_0^2 \mathcal{L}_0 (1 - r_1)^\alpha (1 - r_2)^\alpha \\
&= P_0 \mathcal{L}(\mathcal{L}_0, r_1 + r_2 - r_1 r_2) \\
&= P_0 \mathcal{L}_{\text{singlestep}}
\end{aligned}
$$

Hence, our functional form attains proportional performance between single-step and multi-step pruning for iterative methods.

## C  DATASET DESCRIPTIONS

The WikiText corpus (Merity et al., 2016) is a standard benchmark for language modeling, consisting of human-curated articles that are stylistically polished, factually reliable, and written from a neutral perspective. Two variants exist – WikiText-2 and WikiText-103,with our experiments making use of WikiText-2. The LAMBADA dataset (*Language Modeling Broadened to Account for Discourse Aspects*) (Paperno et al., 2016) contains narrative passages designed to test a model's ability to predict the final word, a task that requires integrating broad context and maintaining discourse-level coherence. PiQA (Bisk et al., 2020) evaluates physical common-sense reasoning in everyday situations, often with unconventional but practical solutions. Each instance poses an instruction-based problem (e.g., how to construct, bake, or manipulate objects) and provides two candidate solutions, of which exactly one is correct, framing the task as multiple-choice QA. WinoGrande (Sakaguchi et al., 2021), derived from the Winograd Schema Challenge (Levesque et al., 2012), is a large-scale benchmark for pronoun resolution, a task trivial for humans yet challenging for AI systems. HellaSwag (Zellers et al., 2019) focuses on common-sense natural language inference: given a context, models must select the most plausible continuation. The AI2 Reasoning Challenge (Clark et al., 2018) provides grade-school science exam questions that require both knowledge and reasoning to solve. Finally, CoQA (Reddy et al., 2019) is a conversational QA dataset containing 127k question-answer pairs across 8k dialogues spanning seven domains. Its questions are conversational in style, and answers are free-form text supported by highlighted evidence in the passage, making it a benchmark for building dialogue-based reading comprehension systems.

# D  RESULTS

## D.1  POST-PRUNING PERFORMANCE OF LLMS

| Model | Reasoning | QA | Language | Average |
|---|---|---|---|---|
| OPT-2.7B | 0.52 | 0.51 | 0.49 | 0.51 |
| OPT-6.7B | 0.57 | 0.55 | 0.54 | 0.56 |
| OPT-13B | 0.58 | 0.56 | 0.56 | 0.57 |
| LLaMA-7B | 0.64 | 0.64 | 0.64 | 0.64 |
| LLaMA-13B | 0.67 | 0.66 | 0.69 | 0.68 |
| Llama-3.1-8B† | 0.69 | 0.68 | 0.69 | 0.69 |
| Phi-3-mini-4k-instruct† | 0.74 | 0.68 | 0.53 | 0.68 |

Table 4: Results obtained by the unpruned models on different tasks. Models highlighted with † are not used to fit the pruning laws, but to test them in an out-of-distribution setup.

In this section, we present the detailed task-level results for both unpruned and pruned LLMs. These results complement the main paper by offering a fine-grained view of performance degradation under pruning, highlighting differences across architectures, pruning ratios, and downstream tasks.

Table 4 reports the accuracy of seven unpruned models across reasoning, QA, and language modeling tasks. As expected, performance generally improves with scale: OPT-2.7B achieves an average of $0.51$, whereas OPT-13B and LLaMA-13B improve to $0.57$ and $0.68$, respectively. The LLaMA series consistently outperforms OPT at similar parameter scales, underscoring architectural and training advantages. Notably, Phi-3-Mini-4K-Instruct matches the average of LLaMA-13B ($0.68$) despite being smaller in size, and LLaMA-3.1-8B surpasses both ($0.69$). These results confirm that newer families, even at comparable or smaller parameter counts, can rival or outperform older architectures. These baseline numbers provide the $\mathcal{L}_0$ values against which pruning performance is evaluated.

**Performance under pruning.** Table 5 reports accuracy across pruning ratios from $0.1$ to $0.9$. Several consistent patterns emerge:

- **Early pruning robustness.** At pruning ratios up to $0.3$, most models retain a majority of their baseline accuracy. For example, LLaMA-13B maintains an average of $0.49 \pm 0.19$ at pruning ratio $0.3$, compared to its baseline $0.68$, suggesting a gradual initial degradation. Similar behavior is observed for LLaMA-7B ($0.52 \pm 0.12$ at ratio $0.3$ vs. $0.64$ baseline).
- **Critical threshold around 0.5.** Across all models, pruning ratios around $0.5$ mark a sharp turning point. For instance, OPT-2.7B drops from $0.41 \pm 0.11$ at ratio $0.3$ to $0.35 \pm 0.13$ at ratio $0.5$, and LLaMA-13B falls to $0.41 \pm 0.22$. This regime represents the "critical pruning threshold", beyond which accuracy begins to collapse.
- **Collapse at extreme pruning.** At pruning ratios of $0.8$ and higher, models largely fail across tasks, with QA accuracies dropping close to zero. OPT-13B at $0.9$ achieves $0.0 \pm 0.0$ in QA, while LLaMA-7B maintains only $0.26 \pm 0.03$ on average. These near-zero values highlight the infeasibility of aggressive pruning without recovery fine-tuning.

**Task-level differences.** Task sensitivity to pruning is heterogeneous. Reasoning tasks are consistently more robust than QA or language modeling. For example, at pruning ratio $0.5$, LLaMA-13B retains $0.49 \pm 0.16$ in reasoning, compared to $0.26 \pm 0.36$ in QA and $0.30 \pm 0.28$ in language modeling. This robustness of reasoning tasks aligns with earlier findings that such benchmarks rely on distributed knowledge across layers, making them less vulnerable to pruning entire components. By contrast, QA tasks exhibit high variance and rapid collapse, with standard deviations reaching $0.31$ (LLaMA-7B at ratio $0.4$). Language modeling sits in between: degradation is faster than reasoning but less volatile than QA.

**Variance analysis.** A notable observation is the increase in standard deviation as pruning ratio increases, particularly for QA tasks. For instance, OPT-13B exhibits a deviation of $0.28$ at pruning ratio $0.4$ for QA, compared to just $0.16$ at ratio $0.2$. This indicates that pruning disproportionately destabilizes QA performance, making it harder to predict with certainty. Conversely, reasoning

accuracies exhibit smaller standard errors (often $\leq 0.1$), suggesting greater consistency. This variance structure reinforces the importance of task-aware pruning strategies.

**Model family comparisons.** Across families, LLaMA models consistently outperform OPT at the same pruning ratios, retaining higher averages and showing smoother degradation curves. At pruning ratio 0.5, LLaMA-7B maintains $0.41 \pm 0.19$, while OPT-6.7B drops to $0.35 \pm 0.14$. Larger models in both families show slightly better resilience than smaller ones, but the advantage diminishes beyond 0.5, where both collapse rapidly. This suggests that scaling provides robustness only up to moderate pruning ratios.

| Model | Pruning Ratio | QA | Reasoning | Language | Average |
|---|---|---|---|---|---|
| | 0.1 | $0.51 \pm 0.22$ | $0.64 \pm 0.12$ | $0.46 \pm 0.25$ | $0.58 \pm 0.16$ |
| | 0.2 | $0.41 \pm 0.31$ | $0.59 \pm 0.14$ | $0.4 \pm 0.27$ | $0.52 \pm 0.2$ |
| | 0.3 | $0.38 \pm 0.31$ | $0.6 \pm 0.1$ | $0.39 \pm 0.24$ | $0.49 \pm 0.19$ |
| | 0.4 | $0.34 \pm 0.31$ | $0.57 \pm 0.12$ | $0.36 \pm 0.25$ | $0.46 \pm 0.2$ |
| LLaMA-13B | 0.5 | $0.26 \pm 0.36$ | $0.49 \pm 0.16$ | $0.3 \pm 0.28$ | $0.41 \pm 0.22$ |
| | 0.6 | $0.21 \pm 0.34$ | $0.46 \pm 0.14$ | $0.27 \pm 0.24$ | $0.38 \pm 0.19$ |
| | 0.7 | $0.15 \pm 0.25$ | $0.41 \pm 0.07$ | $0.18 \pm 0.12$ | $0.32 \pm 0.1$ |
| | 0.8 | $0.01 \pm 0.02$ | $0.37 \pm 0.01$ | $0.13 \pm 0.04$ | $0.27 \pm 0.01$ |
| | 0.9 | $0.0 \pm 0.0$ | $0.37 \pm 0.01$ | $0.09 \pm 0.01$ | $0.25 \pm 0.01$ |
| | 0.1 | $0.49 \pm 0.15$ | $0.61 \pm 0.08$ | $0.43 \pm 0.22$ | $0.55 \pm 0.13$ |
| | 0.2 | $0.49 \pm 0.14$ | $0.6 \pm 0.08$ | $0.41 \pm 0.2$ | $0.54 \pm 0.12$ |
| | 0.3 | $0.48 \pm 0.16$ | $0.59 \pm 0.09$ | $0.38 \pm 0.19$ | $0.52 \pm 0.12$ |
| | 0.4 | $0.29 \pm 0.31$ | $0.51 \pm 0.14$ | $0.31 \pm 0.24$ | $0.43 \pm 0.19$ |
| LLaMA-7B | 0.5 | $0.24 \pm 0.32$ | $0.48 \pm 0.14$ | $0.29 \pm 0.23$ | $0.41 \pm 0.19$ |
| | 0.6 | $0.21 \pm 0.33$ | $0.46 \pm 0.12$ | $0.25 \pm 0.18$ | $0.37 \pm 0.16$ |
| | 0.7 | $0.11 \pm 0.15$ | $0.4 \pm 0.04$ | $0.18 \pm 0.08$ | $0.31 \pm 0.06$ |
| | 0.8 | $0.03 \pm 0.03$ | $0.38 \pm 0.03$ | $0.13 \pm 0.03$ | $0.27 \pm 0.02$ |
| | 0.9 | $0.01 \pm 0.02$ | $0.38 \pm 0.03$ | $0.1 \pm 0.03$ | $0.26 \pm 0.03$ |
| | 0.1 | $0.37 \pm 0.32$ | $0.54 \pm 0.14$ | $0.39 \pm 0.26$ | $0.48 \pm 0.19$ |
| | 0.2 | $0.35 \pm 0.3$ | $0.53 \pm 0.14$ | $0.36 \pm 0.25$ | $0.46 \pm 0.19$ |
| | 0.3 | $0.32 \pm 0.28$ | $0.51 \pm 0.14$ | $0.33 \pm 0.24$ | $0.44 \pm 0.18$ |
| | 0.4 | $0.28 \pm 0.27$ | $0.5 \pm 0.13$ | $0.31 \pm 0.23$ | $0.42 \pm 0.17$ |
| OPT-13B | 0.5 | $0.22 \pm 0.25$ | $0.47 \pm 0.12$ | $0.28 \pm 0.22$ | $0.39 \pm 0.16$ |
| | 0.6 | $0.17 \pm 0.26$ | $0.45 \pm 0.11$ | $0.24 \pm 0.21$ | $0.36 \pm 0.15$ |
| | 0.7 | $0.13 \pm 0.21$ | $0.41 \pm 0.08$ | $0.2 \pm 0.15$ | $0.32 \pm 0.11$ |
| | 0.8 | $0.01 \pm 0.02$ | $0.37 \pm 0.03$ | $0.13 \pm 0.07$ | $0.27 \pm 0.04$ |
| | 0.9 | $0.0 \pm 0.0$ | $0.36 \pm 0.0$ | $0.07 \pm 0.01$ | $0.24 \pm 0.0$ |
| | 0.1 | $0.31 \pm 0.26$ | $0.51 \pm 0.08$ | $0.34 \pm 0.19$ | $0.44 \pm 0.13$ |
| | 0.2 | $0.27 \pm 0.23$ | $0.5 \pm 0.07$ | $0.32 \pm 0.17$ | $0.43 \pm 0.11$ |
| | 0.3 | $0.2 \pm 0.25$ | $0.49 \pm 0.06$ | $0.3 \pm 0.16$ | $0.41 \pm 0.11$ |
| | 0.4 | $0.16 \pm 0.25$ | $0.46 \pm 0.08$ | $0.27 \pm 0.17$ | $0.37 \pm 0.13$ |
| OPT-2.7B | 0.5 | $0.14 \pm 0.23$ | $0.43 \pm 0.09$ | $0.24 \pm 0.18$ | $0.35 \pm 0.13$ |
| | 0.6 | $0.12 \pm 0.19$ | $0.42 \pm 0.08$ | $0.21 \pm 0.15$ | $0.33 \pm 0.11$ |
| | 0.7 | $0.05 \pm 0.09$ | $0.39 \pm 0.05$ | $0.16 \pm 0.1$ | $0.29 \pm 0.07$ |
| | 0.8 | $0.02 \pm 0.03$ | $0.37 \pm 0.01$ | $0.12 \pm 0.05$ | $0.26 \pm 0.02$ |
| | 0.9 | $0.0 \pm 0.0$ | $0.36 \pm 0.01$ | $0.09 \pm 0.01$ | $0.25 \pm 0.01$ |
| | 0.1 | $0.52 \pm 0.04$ | $0.6 \pm 0.01$ | $0.49 \pm 0.05$ | $0.56 \pm 0.02$ |
| | 0.2 | $0.35 \pm 0.26$ | $0.54 \pm 0.08$ | $0.38 \pm 0.17$ | $0.48 \pm 0.12$ |
| | 0.3 | $0.13 \pm 0.18$ | $0.45 \pm 0.11$ | $0.22 \pm 0.17$ | $0.36 \pm 0.13$ |
| | 0.4 | $0.21 \pm 0.27$ | $0.48 \pm 0.11$ | $0.31 \pm 0.21$ | $0.41 \pm 0.15$ |
| OPT-6.7B | 0.5 | $0.17 \pm 0.28$ | $0.46 \pm 0.11$ | $0.28 \pm 0.22$ | $0.38 \pm 0.16$ |
| | 0.6 | $0.15 \pm 0.25$ | $0.43 \pm 0.1$ | $0.24 \pm 0.2$ | $0.35 \pm 0.14$ |
| | 0.7 | $0.1 \pm 0.18$ | $0.4 \pm 0.07$ | $0.19 \pm 0.14$ | $0.31 \pm 0.1$ |
| | 0.8 | $0.01 \pm 0.02$ | $0.37 \pm 0.02$ | $0.12 \pm 0.06$ | $0.26 \pm 0.03$ |
| | 0.9 | $0.0 \pm 0.0$ | $0.36 \pm 0.0$ | $0.08 \pm 0.01$ | $0.24 \pm 0.0$ |

Table 5: Average performance of different pruned LLMs for different pruning strategies.

## D.2 IMPACT OF PRUNING ON INFERENCE EFFICIENCY OF LLMs

We now report detailed results on inference efficiency of pruned LLMs, measured as relative speedup over the unpruned baseline (Table 6). These values complement the performance numbers in Table 5, offering a comprehensive view of the accuracy-efficiency trade-offs introduced by pruning.

**Overall trends.** Inference speedup increases consistently with higher pruning ratios, but the extent of improvement is highly dependent on the pruning method. Depth pruning yields the largest runtime

| Model | Pruning Ratio | DepthPruning | Unstructured | WidthPruning |
|---|---|---|---|---|
| | 0.1 | 1.42 | 1.15 | 0.98 |
| | 0.2 | 1.52 | 1.15 | 1.04 |
| | 0.3 | 1.60 | 1.15 | 1.15 |
| | 0.4 | 1.72 | 1.16 | 1.19 |
| LLaMA-13B | 0.5 | 1.84 | 1.16 | 1.15 |
| | 0.6 | 1.84 | 1.17 | 1.28 |
| | 0.7 | 1.84 | 1.18 | 1.36 |
| | 0.8 | 1.84 | 1.18 | 1.44 |
| | 0.9 | 1.84 | 1.19 | 1.36 |
| | 0.1 | 1.34 | 1.24 | 1.00 |
| | 0.2 | 0.85 | 1.24 | 0.94 |
| | 0.3 | 1.28 | 1.24 | 1.00 |
| | 0.4 | 1.63 | 1.24 | 0.94 |
| LLaMA-7B | 0.5 | 1.63 | 1.25 | 1.02 |
| | 0.6 | 1.63 | 1.25 | 1.05 |
| | 0.7 | 1.63 | 1.25 | 1.01 |
| | 0.8 | 1.63 | 1.25 | 1.05 |
| | 0.9 | 1.63 | 1.26 | 0.98 |
| | 0.1 | 1.07 | 1.22 | 0.90 |
| | 0.2 | 1.20 | 1.22 | 0.98 |
| | 0.3 | 1.35 | 1.22 | 0.98 |
| | 0.4 | 1.54 | 1.23 | 1.11 |
| OPT-13B | 0.5 | 1.80 | 1.24 | 1.10 |
| | 0.6 | 2.13 | 1.24 | 1.17 |
| | 0.7 | 2.74 | 1.24 | 1.26 |
| | 0.8 | 3.51 | 1.25 | 1.37 |
| | 0.9 | 5.05 | 1.26 | 1.18 |
| | 0.1 | 1.40 | 1.29 | 0.90 |
| | 0.2 | 1.30 | 1.30 | 0.82 |
| | 0.3 | 1.29 | 1.29 | 0.86 |
| | 0.4 | 1.43 | 1.29 | 0.88 |
| OPT-2.7B | 0.5 | 1.75 | 1.29 | 0.91 |
| | 0.6 | 1.75 | 1.30 | 0.84 |
| | 0.7 | 1.75 | 1.31 | 0.90 |
| | 0.8 | 1.75 | 1.30 | 0.87 |
| | 0.9 | 1.75 | 1.31 | 0.86 |
| | 0.1 | 1.07 | 1.19 | 0.94 |
| | 0.2 | 1.17 | 1.20 | 0.88 |
| | 0.3 | 1.32 | 1.19 | 0.93 |
| | 0.4 | 1.49 | 1.20 | 0.98 |
| OPT-6.7B | 0.5 | 1.79 | 1.20 | 1.02 |
| | 0.6 | 2.13 | 1.20 | 1.01 |
| | 0.7 | 2.63 | 1.21 | 1.08 |
| | 0.8 | 3.44 | 1.21 | 1.08 |
| | 0.9 | 5.01 | 1.22 | 1.04 |

Table 6: Inference speedup over unpruned LLMs for different pruning ratios.

benefits, with speedups ranging from $1.4\times$ at 10% pruning to more than $5\times$ at 90% pruning for OPT-13B and OPT-6.7B. In contrast, unstructured pruning offers little to no improvement (typically between $1.15\times$ and $1.30\times$) because current hardware and software stacks do not efficiently exploit sparsity without specialized kernels. Width pruning provides a balanced trade-off: modest but consistent gains between $1.0\times$ and $1.4\times$, reflecting the structured nature of the removed components.

**Model-level differences.** Model scale strongly affects achievable speedup. Larger models such as OPT-13B and OPT-6.7B benefit disproportionately from depth pruning: OPT-13B achieves $2.13\times$ at 60% pruning and $5.05\times$ at 90%, while OPT-6.7B follows a nearly identical trend with a peak of $5.01\times$ at 90%. By contrast, smaller models saturate earlier. For example, OPT-2.7B reaches only $1.75\times$ even at pruning ratios $\geq 0.5$, and LLaMA-7B plateaus at $1.63\times$ beyond 40% pruning. This indicates diminishing efficiency returns when pruning models below $\sim$ 7B parameters. LLaMA-13B lies between these extremes, achieving $1.84\times$ at 50-90% pruning, reflecting both its scale and architecture.

**Method-level differences.** The method comparison further clarifies trade-offs.

- **Depth pruning** is the most effective for runtime efficiency, as entire Transformer blocks are removed, reducing both FLOPs and memory. Its benefits scale super-linearly with pruning ratio, particularly in OPT models.
- **Unstructured pruning** achieves minimal speedup (e.g., $1.17\times$ for LLaMA-13B at 60%), since irregular sparsity patterns hinder hardware acceleration. These results suggest unstructured pruning should be preferred only when accuracy preservation is prioritized.
- **Width pruning** provides moderate efficiency gains (up to $1.44\times$ for LLaMA-13B and $1.37\times$ for OPT-13B), balancing performance preservation with practical acceleration.

## D.3 Detailed Analysis of Fitted Pruning Laws

| Task | Model | $\alpha$ | $P_0$ | Adj $R^2$ | F Statistic | Test Error |
|------|-------|----------|-------|-----------|-------------|------------|
| **QA** | OPT-2.7B | $2.47 \pm 0.24$ | $1.19 \pm 0.27$ | 0.93 | 106.36 | 0.02 |
| **Reasoning** | OPT-2.7B | $0.17 \pm 0.02$ | $0.96 \pm 0.02$ | 0.89 | 65.37 | 0.04 |
| **Language** | OPT-2.7B | $0.65 \pm 0.02$ | $0.75 \pm 0.02$ | 0.99 | 1268.91 | 0.01 |
| **Average** | OPT-2.7B | $0.28 \pm 0.03$ | $0.85 \pm 0.03$ | 0.93 | 109.53 | 0.03 |
| **QA** | OPT-6.7B | $2.6 \pm 0.33$ | $1.44 \pm 0.37$ | 0.88 | 61.1 | 0.05 |
| **Reasoning** | OPT-6.7B | $0.21 \pm 0.04$ | $0.96 \pm 0.04$ | 0.82 | 36.48 | 0.11 |
| **Language** | OPT-6.7B | $0.74 \pm 0.09$ | $0.8 \pm 0.1$ | 0.9 | 73.67 | 0.06 |
| **Average** | OPT-6.7B | $0.34 \pm 0.05$ | $0.87 \pm 0.06$ | 0.83 | 39.88 | 0.1 |
| **QA** | LLaMA-7B | $1.84 \pm 0.12$ | $1.24 \pm 0.13$ | 0.97 | 243.24 | 0.1 |
| **Reasoning** | LLaMA-7B | $0.25 \pm 0.04$ | $0.93 \pm 0.04$ | 0.85 | 47.06 | 0.08 |
| **Language** | LLaMA-7B | $0.71 \pm 0.04$ | $0.72 \pm 0.04$ | 0.98 | 383.52 | 0.03 |
| **Average** | LLaMA-7B | $0.38 \pm 0.04$ | $0.86 \pm 0.05$ | 0.9 | 74.55 | 0.07 |
| **QA** | OPT-13B | $2.62 \pm 0.33$ | $1.76 \pm 0.37$ | 0.88 | 61.79 | 0.06 |
| **Reasoning** | OPT-13B | $0.21 \pm 0.01$ | $0.94 \pm 0.02$ | 0.96 | 212.27 | 0.02 |
| **Language** | OPT-13B | $0.75 \pm 0.03$ | $0.8 \pm 0.03$ | 0.99 | 696.71 | 0.02 |
| **Average** | OPT-13B | $0.34 \pm 0.02$ | $0.87 \pm 0.02$ | 0.98 | 375.15 | 0.02 |
| **QA** | LLaMA-13B | $2.57 \pm 0.31$ | $1.74 \pm 0.34$ | 0.9 | 70.56 | 0.05 |
| **Reasoning** | LLaMA-13B | $0.28 \pm 0.04$ | $0.93 \pm 0.04$ | 0.88 | 61.4 | 0.07 |
| **Language** | LLaMA-13B | $0.78 \pm 0.04$ | $0.73 \pm 0.04$ | 0.98 | 492.22 | 0.03 |
| **Average** | LLaMA-13B | $0.4 \pm 0.04$ | $0.83 \pm 0.04$ | 0.94 | 126.08 | 0.05 |

Table 7: Coefficients of the model-task-level pruning laws.

We now analyze the coefficients of the pruning laws fitted across different levels of aggregation: model-task (Table 7), method-task (Table 8), and method-model-tasks (Table 9). Recall that our pruning law takes the form:

$$\mathcal{L}_1 = \mathcal{L}_0(1-r)^\alpha P_0,$$

where $\mathcal{L}_0$ is the unpruned model performance, $r$ is the pruning ratio, $\alpha$ governs the sensitivity to the retention ratio $(1-r)$, and $P_0$ captures the bias term reflecting the expected loss offset after pruning.

| Task | Method | $\alpha$ | $P_0$ | Adj $R^2$ | F Statistic | Test Error |
|------|--------|----------|-------|-----------|-------------|------------|
| QA | Unstructured | $2.71 \pm 0.2$ | $3.56 \pm 0.22$ | 0.81 | 183.76 | 0.25 |
| Reasoning | Unstructured | $0.31 \pm 0.02$ | $1.17 \pm 0.02$ | 0.85 | 252.22 | 0.12 |
| Language | Unstructured | $0.91 \pm 0.04$ | $1.39 \pm 0.05$ | 0.91 | 418.66 | 0.18 |
| Average | Unstructured | $0.47 \pm 0.03$ | $1.22 \pm 0.03$ | 0.90 | 370.20 | 0.15 |
| QA | DepthPruning | $1.42 \pm 0.43$ | $0.03 \pm 0.48$ | 0.18 | 10.70 | 0.02 |
| Reasoning | DepthPruning | $0.1 \pm 0.03$ | $0.74 \pm 0.03$ | 0.22 | 13.11 | 0.09 |
| Language | DepthPruning | $0.28 \pm 0.07$ | $0.25 \pm 0.08$ | 0.23 | 14.10 | 0.05 |
| Average | DepthPruning | $0.13 \pm 0.04$ | $0.54 \pm 0.04$ | 0.20 | 12.21 | 0.10 |
| QA | WidthPruning | $3.25 \pm 0.21$ | $0.99 \pm 0.23$ | 0.84 | 237.08 | 0.03 |
| Reasoning | WidthPruning | $0.26 \pm 0.03$ | $0.93 \pm 0.03$ | 0.62 | 72.91 | 0.11 |
| Language | WidthPruning | $0.82 \pm 0.05$ | $0.7 \pm 0.06$ | 0.86 | 262.08 | 0.05 |
| Average | WidthPruning | $0.39 \pm 0.04$ | $0.83 \pm 0.04$ | 0.70 | 103.11 | 0.10 |

Table 8: Coefficients of the method-task-level pruning laws.

**Model-task level coefficients.**    Table 7 highlights consistent differences across tasks and model scales. For QA, $\alpha$ values are notably large (e.g., OPT-2.7B: $\alpha = 2.47 \pm 0.24$, OPT-13B: $\alpha = 2.62 \pm 0.33$), indicating steep degradation as pruning increases. In contrast, reasoning tasks show very small $\alpha$ values across all models (ranging from 0.17-0.28), reflecting robustness to pruning. Language modeling tasks fall in between ($\alpha \approx$ 0.65-0.78), consistent with their moderate sensitivity.

Bias terms $P_0$ also exhibit systematic variation: reasoning tasks have $P_0 \approx$ 0.93-0.96, meaning their baseline is well preserved even at moderate pruning. QA tasks have much higher $P_0$ (e.g., $1.74 \pm 0.34$ for LLaMA-13B), amplifying their initial performance before the steep decline governed by $\alpha$. High $P_0$ coupled with large $\alpha$ values suggests QA tasks are brittle: they initially retain performance but collapse quickly beyond the critical pruning ratio. These findings corroborate the raw results in Table 5, where reasoning tasks degraded more gracefully than QA or language modeling.

**Method-task level coefficients.**    Table 8 reports coefficients averaged over models, disaggregated by pruning method. Unstructured pruning exhibits the largest $\alpha$ across tasks ($\alpha = 2.71 \pm 0.20$ for QA, $0.91 \pm 0.04$ for language), reflecting higher sensitivity to pruning, but it also shows very high $P_0$ values (e.g., $3.56 \pm 0.22$ for QA). This indicates that while unstructured pruning begins with inflated performance retention, it rapidly declines with higher ratios. Depth pruning has very small $\alpha$ values across all tasks ($\alpha = $ 0.1-0.28), and $P_0$ values near or below 1.0, confirming that it delivers consistent but more linear degradation patterns. Width pruning shows the most aggressive $\alpha$ for QA ($3.25 \pm 0.21$) but more moderate values for reasoning and language, combined with $P_0 \approx$ 0.7-0.99. This suggests width pruning offers a middle ground, its degradation is task-sensitive but more balanced than unstructured pruning.

**Method-model-task level coefficients.**    The finest-grained analysis (Table 9) reveals interactions between model scale, pruning method, and task. Several patterns are evident:

- **Small models are more brittle.** For OPT-2.7B under unstructured pruning, QA $\alpha$ values exceed 2.6, with $P_0 \approx 3.08$, while reasoning remains near $\alpha = 0.23$. This implies extreme instability in QA, contrasting with stable reasoning.
- **Large models degrade more smoothly.** For LLaMA-13B under width pruning, QA has $\alpha = 3.06 \pm 0.46$ but $P_0 = 1.22 \pm 0.51$, while reasoning is closer to $\alpha = 0.29 \pm 0.07$. The gap between QA and reasoning persists, but higher $P_0$ ensures smoother average degradation relative to smaller OPT models.
- **Depth pruning produces flatter curves.** Across all models, $\alpha$ remains small ($< 0.3$), and $P_0$ tends to be $< 1.0$, producing slow, near-linear degradation with pruning ratio. This validates the efficiency advantages of depth pruning discussed in the previous section.

**Goodness of fit.**    Adjusted $R^2$ values are consistently high for unstructured and width pruning (0.7-0.99), demonstrating excellent explanatory power. Depth pruning fits are weaker (often $R^2 < 0.3$), reflecting the flatter degradation curves and smaller dynamic range. F-statistics confirm statistical significance across most settings, and test errors remain small (typically 0.02-0.15), indicating strong

| Task | Model | Method | $\alpha$ | $P_0$ | Adj $R^2$ | F Statistic | Test Error |
|---|---|---|---|---|---|---|---|
| QA | | | $2.63 \pm 0.42$ | $3.08 \pm 0.46$ | 0.83 | 39.87 | 0.15 |
| Reasoning | OPT-2.7B | | $0.23 \pm 0.02$ | $1.15 \pm 0.03$ | 0.92 | 95.53 | 0.07 |
| Language | | | $0.8 \pm 0.07$ | $1.32 \pm 0.08$ | 0.93 | 114.16 | 0.13 |
| Average | | | $0.39 \pm 0.03$ | $1.17 \pm 0.04$ | 0.95 | 139.41 | 0.09 |
| QA | | | $2.77 \pm 0.53$ | $3.82 \pm 0.62$ | 0.79 | 27.24 | 0.23 |
| Reasoning | OPT-6.7B | | $0.27 \pm 0.03$ | $1.17 \pm 0.03$ | 0.94 | 103.33 | 0.08 |
| Language | | | $0.87 \pm 0.12$ | $1.44 \pm 0.14$ | 0.88 | 51.50 | 0.19 |
| Average | | | $0.44 \pm 0.05$ | $1.22 \pm 0.06$ | 0.91 | 76.15 | 0.13 |
| QA | | | $2.71 \pm 0.49$ | $3.65 \pm 0.55$ | 0.78 | 30.05 | 0.31 |
| Reasoning | LLaMA-7B | Unstructured | $0.37 \pm 0.05$ | $1.18 \pm 0.05$ | 0.88 | 59.14 | 0.14 |
| Language | | | $0.95 \pm 0.08$ | $1.34 \pm 0.09$ | 0.95 | 139.71 | 0.16 |
| Average | | | $0.53 \pm 0.06$ | $1.22 \pm 0.07$ | 0.91 | 77.53 | 0.17 |
| QA | | | $2.65 \pm 0.51$ | $3.5 \pm 0.56$ | 0.77 | 27.53 | 0.24 |
| Reasoning | OPT-13B | | $0.28 \pm 0.03$ | $1.18 \pm 0.03$ | 0.93 | 114.94 | 0.10 |
| Language | | | $0.94 \pm 0.13$ | $1.47 \pm 0.14$ | 0.86 | 52.26 | 0.19 |
| Average | | | $0.45 \pm 0.05$ | $1.23 \pm 0.05$ | 0.92 | 94.66 | 0.14 |
| QA | | | $2.77 \pm 0.53$ | $3.87 \pm 0.58$ | 0.77 | 27.73 | 0.36 |
| Reasoning | LLaMA-13B | | $0.38 \pm 0.05$ | $1.19 \pm 0.05$ | 0.89 | 66.52 | 0.15 |
| Language | | | $0.98 \pm 0.1$ | $1.4 \pm 0.11$ | 0.93 | 102.91 | 0.20 |
| Average | | | $0.55 \pm 0.06$ | $1.25 \pm 0.07$ | 0.90 | 73.61 | 0.19 |
| QA | | | $1.51 \pm 0.62$ | $0.02 \pm 0.69$ | 0.38 | 5.85 | 0.06 |
| Reasoning | OPT-2.7B | | $0.06 \pm 0.02$ | $0.79 \pm 0.02$ | 0.43 | 7.03 | 0.06 |
| Language | | | $0.19 \pm 0.07$ | $0.27 \pm 0.08$ | 0.46 | 7.95 | 0.08 |
| Average | | | $0.07 \pm 0.03$ | $0.57 \pm 0.03$ | 0.45 | 7.42 | 0.07 |
| QA | | | $1.8 \pm 1.01$ | $0.03 \pm 1.12$ | 0.21 | 3.18 | 0.00 |
| Reasoning | OPT-6.7B | | $0.14 \pm 0.07$ | $0.78 \pm 0.08$ | 0.24 | 3.51 | 0.18 |
| Language | | | $0.58 \pm 0.23$ | $0.35 \pm 0.26$ | 0.40 | 6.33 | 0.06 |
| Average | | | $0.22 \pm 0.12$ | $0.61 \pm 0.13$ | 0.24 | 3.58 | 0.15 |
| QA | | | $1.2 \pm 0.5$ | $0.33 \pm 0.55$ | 0.37 | 5.78 | 0.18 |
| Reasoning | LLaMA-7B | DepthPruning | $0.13 \pm 0.06$ | $0.78 \pm 0.06$ | 0.35 | 5.40 | 0.15 |
| Language | | | $0.23 \pm 0.11$ | $0.31 \pm 0.12$ | 0.31 | 4.61 | 0.16 |
| Average | | | $0.2 \pm 0.09$ | $0.62 \pm 0.09$ | 0.35 | 5.32 | 0.17 |
| Reasoning | | | $0.01 \pm 0.01$ | $0.63 \pm 0.01$ | 0.15 | 2.37 | 0.04 |
| Language | | | $0.1 \pm 0.06$ | $0.15 \pm 0.06$ | 0.23 | 3.35 | 0.02 |
| Average | | | $0.02 \pm 0.01$ | $0.43 \pm 0.01$ | 0.22 | 3.22 | 0.03 |
| QA | | | $2.59 \pm 0.8$ | $0.11 \pm 0.89$ | 0.54 | 10.45 | 0.01 |
| Reasoning | LLaMA-13B | | $0.15 \pm 0.06$ | $0.73 \pm 0.07$ | 0.37 | 5.72 | 0.15 |
| Language | | | $0.29 \pm 0.12$ | $0.25 \pm 0.13$ | 0.40 | 6.37 | 0.07 |
| Average | | | $0.14 \pm 0.05$ | $0.49 \pm 0.06$ | 0.44 | 7.37 | 0.10 |
| QA | | | $3.0 \pm 0.53$ | $0.42 \pm 0.59$ | 0.80 | 32.30 | 0.01 |
| Reasoning | OPT-2.7B | | $0.2 \pm 0.05$ | $0.95 \pm 0.06$ | 0.64 | 15.03 | 0.11 |
| Language | | | $0.76 \pm 0.1$ | $0.71 \pm 0.11$ | 0.87 | 56.24 | 0.04 |
| Average | | | $0.32 \pm 0.08$ | $0.81 \pm 0.09$ | 0.67 | 17.56 | 0.10 |
| QA | | | $3.37 \pm 0.56$ | $0.85 \pm 0.62$ | 0.82 | 36.58 | 0.03 |
| Reasoning | OPT-6.7B | | $0.26 \pm 0.05$ | $0.99 \pm 0.06$ | 0.77 | 27.21 | 0.08 |
| Language | | | $0.83 \pm 0.08$ | $0.8 \pm 0.08$ | 0.94 | 118.73 | 0.03 |
| Average | | | $0.4 \pm 0.07$ | $0.88 \pm 0.08$ | 0.78 | 30.19 | 0.09 |
| QA | | | $3.33 \pm 0.37$ | $1.1 \pm 0.41$ | 0.91 | 79.60 | 0.03 |
| Reasoning | LLaMA-7B | WidthPruning | $0.25 \pm 0.07$ | $0.84 \pm 0.07$ | 0.63 | 14.34 | 0.14 |
| Language | | | $0.82 \pm 0.09$ | $0.59 \pm 0.1$ | 0.91 | 80.98 | 0.04 |
| Average | | | $0.38 \pm 0.09$ | $0.73 \pm 0.1$ | 0.70 | 19.61 | 0.11 |
| QA | | | $3.48 \pm 0.39$ | $1.91 \pm 0.43$ | 0.91 | 81.04 | 0.06 |
| Reasoning | OPT-13B | | $0.28 \pm 0.04$ | $1.04 \pm 0.05$ | 0.85 | 45.94 | 0.07 |
| Language | | | $0.84 \pm 0.07$ | $0.87 \pm 0.08$ | 0.95 | 151.76 | 0.03 |
| Average | | | $0.43 \pm 0.06$ | $0.97 \pm 0.07$ | 0.85 | 48.01 | 0.07 |
| QA | | | $3.06 \pm 0.46$ | $1.22 \pm 0.51$ | 0.84 | 43.97 | 0.04 |
| Reasoning | LLaMA-13B | | $0.29 \pm 0.07$ | $0.87 \pm 0.08$ | 0.65 | 16.09 | 0.15 |
| Language | | | $0.83 \pm 0.1$ | $0.59 \pm 0.11$ | 0.89 | 66.53 | 0.05 |
| Average | | | $0.43 \pm 0.09$ | $0.77 \pm 0.1$ | 0.72 | 21.24 | 0.11 |

Table 9: Coefficients of method-model-task level pruning laws.

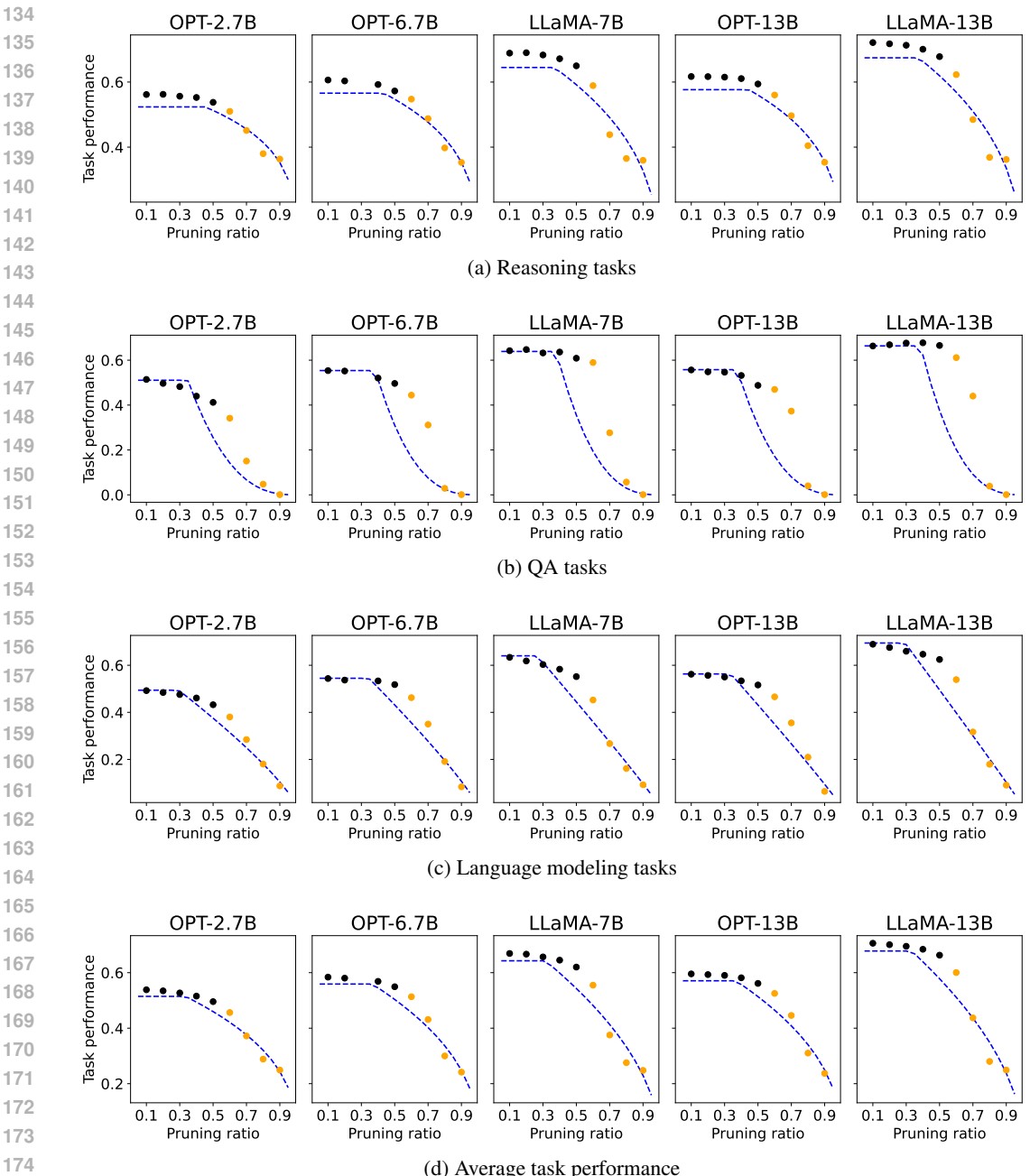

Figure 6: Fitted pruning laws for downstream performance of pruned LLMs with unstructured pruning.

predictive fidelity of the pruning laws. Importantly, even with task and model heterogeneity, the pruning law retains goodness-of-fit across all aggregation levels, supporting its universality.

**Implications.** The fitted coefficients highlight several actionable takeaways:

- **Task dependence.** QA is highly sensitive to pruning ($\alpha > 2$), while reasoning is robust ($\alpha < 0.3$). Practitioners deploying pruned models on QA-heavy benchmarks should adopt conservative pruning ratios or fine-tuning.

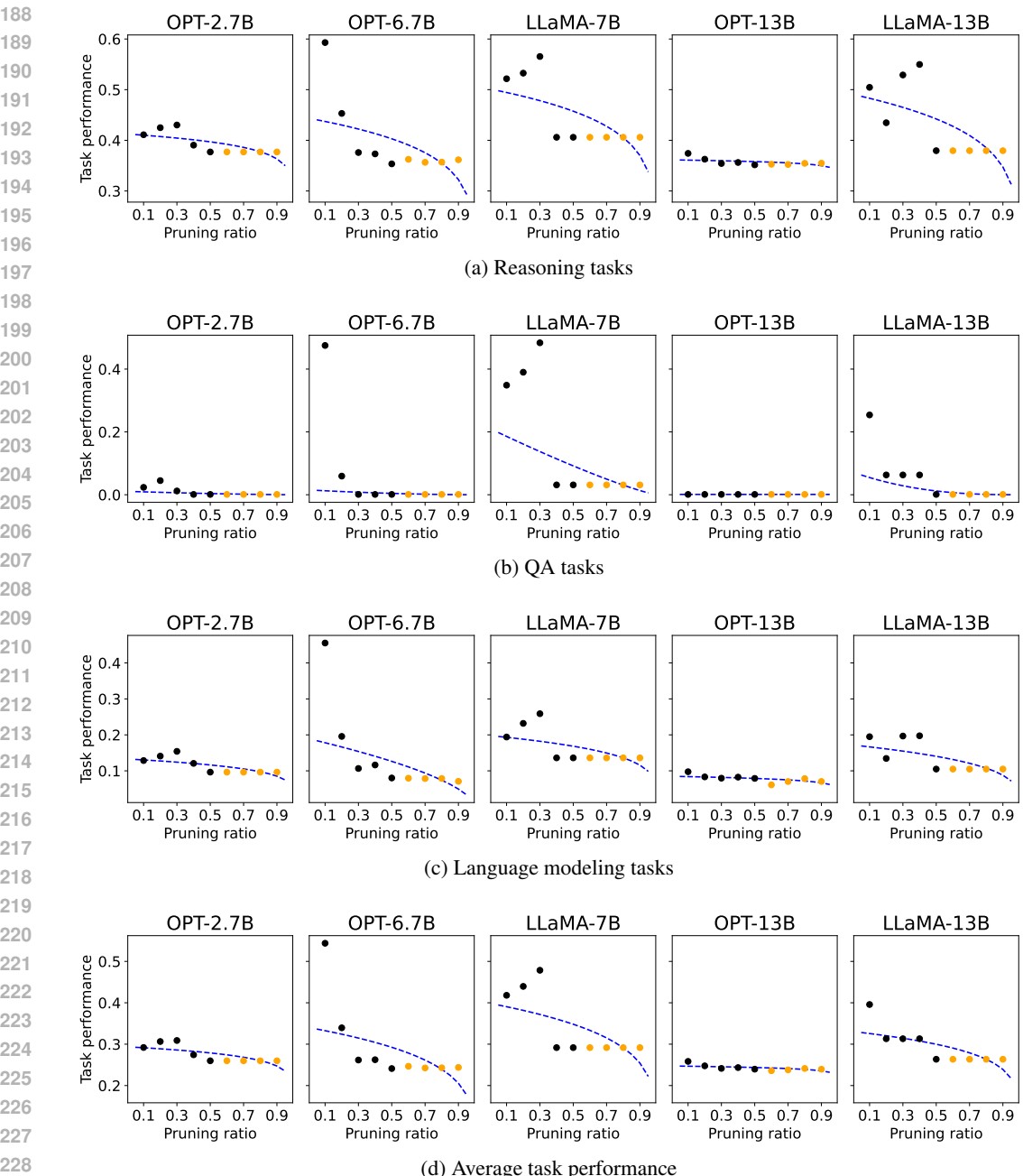

Figure 7: Fitted pruning laws for downstream performance of pruned LLMs with depth pruning.

- **Method choice.** Depth pruning produces the smoothest degradation ($\alpha$ small, $P_0$ near 1), while unstructured pruning exaggerates performance retention ($P_0 \gg 1$) but collapses quickly. Width pruning balances these extremes.
- **Model scale.** Larger models (e.g., LLaMA-13B) show smaller $\alpha$ and smoother degradation compared to smaller OPT models, highlighting that pruning is more reliable for larger LLMs.
- **Universality.** Despite variance across methods and tasks, the pruning law consistently explains observed degradation with low test errors and high $R^2$, confirming its applicability as a general predictive framework.

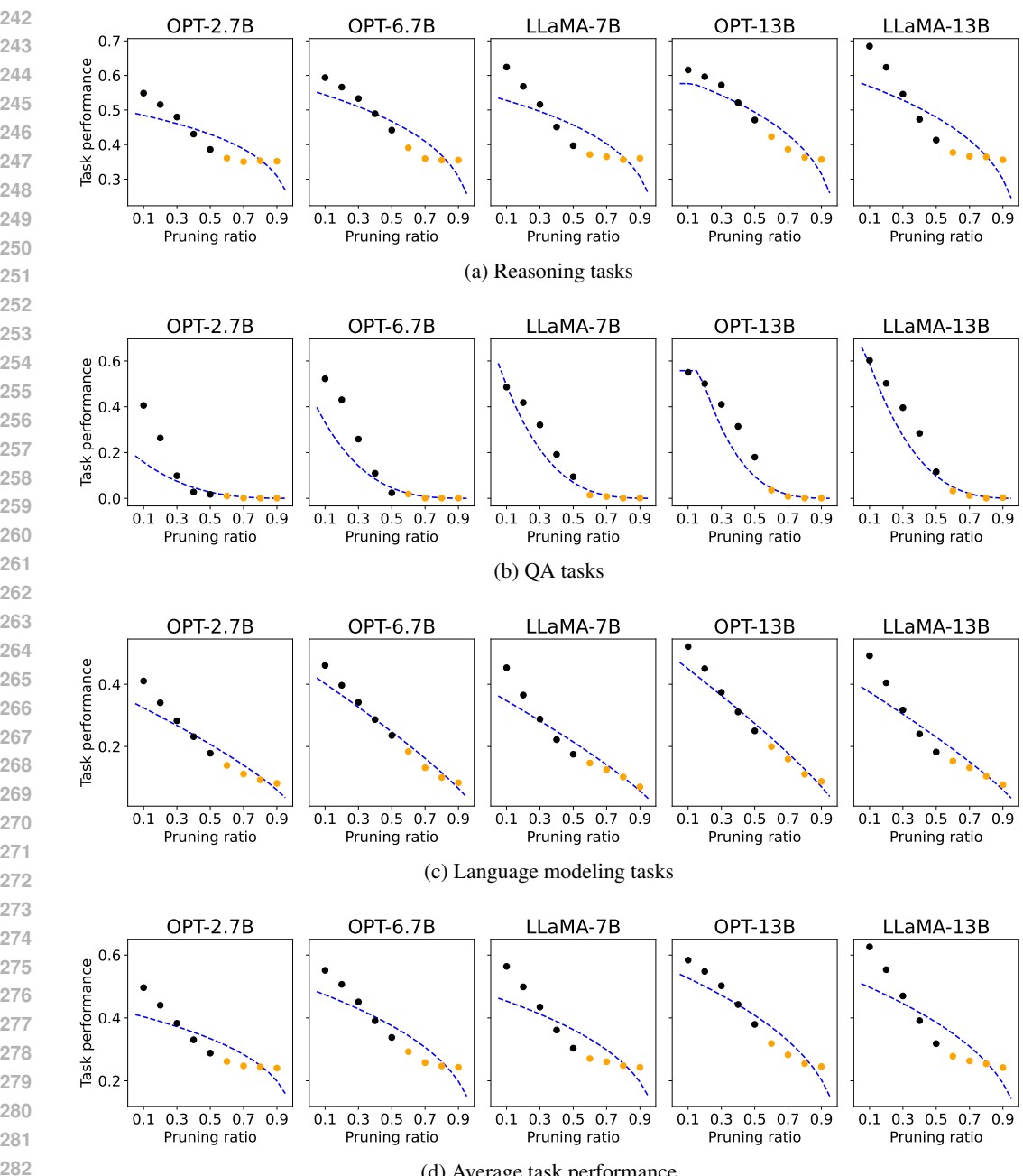

Figure 8: Fitted pruning laws for downstream performance of pruned LLMs with width pruning.

Overall, the fitted coefficients reveal how $\alpha$ and $P_0$ jointly govern pruning dynamics: $\alpha$ determines the slope of decay, while $P_0$ modulates the starting point and task bias. Their complementary roles allow pruning laws to capture a wide variety of behaviors across models, methods, and tasks with a single functional form.

## E    CURIOUS CASE OF KNOWLEDGE EXTRACTION TASKS

In contrast to reasoning, QA, and language modeling, pruning laws fail to capture systematic patterns in knowledge-extraction tasks. We evaluate the pruned LLMs on subsets of MMLU (Hendrycks et al., 2020) knowledge-extraction dataset, including college mathematics, conceptual physics, and global

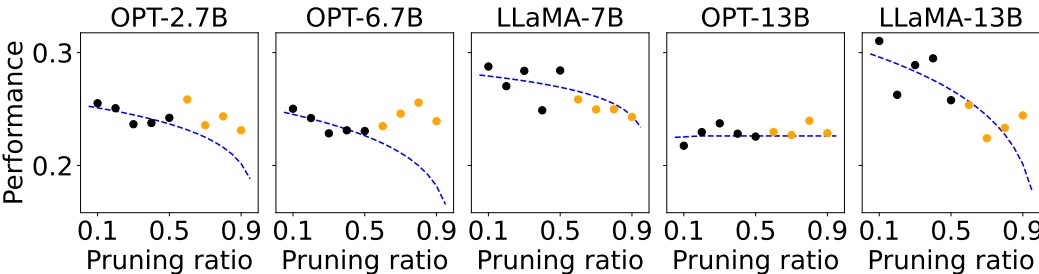

Figure 9: Fitted pruning laws for pruned LLMs on knowledge-extraction tasks. Poor $R^2$ of 0.02 and F-statistic of 1.92, indicate a poor goodness of fit. Similar erratic behavior observed by Sengupta et al. (2025).

facts. As shown in Figure 9, the fitted curves yield extremely poor goodness-of-fit ($R^2 = 0.02$, F-statistic = 1.92), with performance points scattering erratically across pruning ratios. This irregularity likely stems from structural properties of the benchmarks: (i) answers are often revealed directly in the context, allowing even heavily pruned models to rely on surface cues; (ii) the multiple-choice format enables correct guesses from statistical biases unrelated to retained capacity; and (iii) the smooth power-law decay assumed by pruning laws is misaligned with threshold-like behavior in factual recall. Consequently, knowledge tasks do not degrade in a principled manner under pruning and thus offer limited diagnostic value for evaluating universality. While suitable for unpruned models, they should not be used to study compression dynamics; more robust alternatives, such as multi-hop fact retrieval or open-ended factual generation, are needed to probe knowledge retention in pruned LLMs.

## F  PRACTITIONER GUIDELINES FOR SELECTING PRUNING STRATEGIES

To make our findings actionable, we summarize them as practical rules of thumb for real-world adoption of pruning strategies. These guidelines allow practitioners to quickly decide which pruning strategy best matches their constraints and to use the pruning laws for principled planning rather than ad hoc trial-and-error:

1. **Selecting the pruning method:**
   - Use *depth pruning* when runtime speedup is the primary objective, as it yields the largest accelerations (up to $5\times$), though with higher variance in accuracy.
   - Use *unstructured pruning* when performance preservation is critical, since it provides the lowest test errors and the most stable predictions, albeit with modest speedups.
   - Use *width pruning* for a balanced trade-off between accuracy and speed, offering moderate but consistent improvements.

2. **When facing unseen models:**
   - Start with *zero-shot mode* by directly reusing fitted coefficients ($\alpha, P_0$), which achieves low extrapolation error even across families.
   - If limited resources are available to probe the new model, perform *one-shot calibration*: keep $\alpha$ fixed and re-estimate only $P_0$ from a single pruning experiment. This is beneficial when accuracy at specific pruning ratios is critical.

3. **When using new pruning methods:**
   - Begin with zero-shot predictions using coefficients from similar pruning categories.
   - Apply one-shot calibration of $P_0$ if the method is highly different (e.g., algorithmic pruning heuristics not studied in our experiments) and one evaluation run is feasible.

4. **When targeting different tasks:**
   - For *reasoning tasks*, higher pruning ratios can be tolerated, as they are more robust (smaller $\alpha$).
   - For *QA and language modeling*, adopt more conservative pruning ratios, since these tasks degrade faster with pruning.

5. **Compression planning:**
   - If multiple evaluations are possible, test at a few pruning ratios (preferably $< 50\%$) and fit the pruning law locally to refine estimates.
   - If evaluations are infeasible (e.g., resource-constrained scenarios), reuse our published coefficients (model-wise or task-wise) to approximate trade-offs with reasonable accuracy.

In summary, practitioners should use zero-shot pruning laws for fast deployment across new models, tasks, or methods, and fall back to one-shot calibration of $P_0$ when some evaluation budget is available. This ensures a principled, universal, and resource-aware approach to model pruning.

