# OpenReview forum: "Scaling Laws for Parameter Pruning in LLMs"
_ICLR.cc/2026/Conference — ICLR 2026 Conference Withdrawn Submission_

### Official Review · Reviewer_xFt1 · 2025-10-20

**Soundness:** 2
**Presentation:** 2
**Contribution:** 2
**Rating:** 2
**Confidence:** 3

**Summary:**

This paper introduces pruning laws, which are analytical scaling relationships that model how a large language model’s (LLM’s) performance degrades as parameters are pruned. The authors propose a simple power-law formulation $L = L_0 P_0 (1-r)^{\alpha}$ linking post-pruning performance $L$ to base performance $L_0$ and pruning ratio $r$, with fitted parameters $\alpha$ and $P$.

**Strengths:**

1. The experiments are extensive, covering multiple architectures (OPT, LLaMA, Phi-3), pruning strategies, and task types. The universality and cross-method transfer analyses (zero-shot and one-shot setups) convincingly demonstrate robustness.
2. The proposed formulation enables practitioners to estimate safe pruning ratios without retraining or fine-tuning.
3. The derivation of the pruning law, its logarithmic linearization, and the OLS-based fitting procedure are clearly articulated.

**Weaknesses:**

1. The main concern with developing such a scaling law is that post-pruning evaluation is relatively inexpensive compared to pre-training. It is therefore unclear why a separate scaling law is necessary to model post-pruning performance, given that pruning results can typically be obtained within minutes. In addition, AutoML techniques can be employed to efficiently narrow the search space and identify optimal pruning strategies.
2. The study explicitly avoids recovery fine-tuning. While this isolates pruning effects, it limits applicability, since fine-tuning is standard practice for high-quality pruned models. A section quantifying how fine-tuning interacts with the law would be valuable.
3. The proposed pruning method demonstrates limited effectiveness when applied to LLaMA-3.2 and Qwen-3 architectures. In recent work, pruning techniques are often integrated directly into the training process to achieve better efficiency and stability.
4. There are several existing papers on scaling laws for inference-efficient models; it would be helpful if the authors discussed how their approach differs from these works: (1) https://arxiv.org/abs/2401.00448 (2) https://arxiv.org/abs/2501.18107

**Questions:**

See weaknesses.

---

> ### Author Response · Authors · 2025-11-19
> **Response to Reviewer xFt1 Comments - Part I**
>
> We thank the reviewer for providing constructive feedback on our paper. We address the raised concerns below -
>
> ## Motivation of Pruning Laws
>
> While individual post-pruning evaluations are inexpensive, the search space is extremely large: multiple pruning methods and pruning ratios for each model and task of interest. Our results show that performance can vary significantly across these choices, and manual or grid search quickly becomes expensive even if each run is fast.
>
> The value of the pruning law is that it enables:
> - **accurate prediction from very few trials** (3-5 post-pruning evaluations), avoiding dozens of pruning runs;
> - **zero-shot and one-shot prediction** for new models/methods (Table 3), eliminating most of the search entirely;
> - **model- and task-level elasticity coefficients** that let practitioners estimate performance *before* running any pruning.
>
> AutoML still requires many evaluations; the pruning law serves as a **theory-guided prior** that drastically reduces the search space and provides interpretable elasticity parameters. Thus, the law complements (and can even guide) AutoML rather than replacing it.
>
> ## Pruning Laws with RFT
>
> We thank the reviewer for the suggestion. We conduct additional experiments with post-pruning recovery fine-tuning (fine-tuning on the WikiText2 dataset with a subsample size of 1000 and a LoRA rank of 8) and report the results below.
>
> | Task      | $\alpha$        | $P_0$            | Adj $R^2$ | F Statistic | Train Error | Test Error |
> | --------- | ---------------- | ---------------- | --------- | ----------- | ---------- | --------- |
> | Reasoning | 0.24 $\pm$ 0.02 | 0.9 $\pm$ 0.02  | 0.84      | 231.12      | 0.03       | 0.05      |
> | Language  | 0.47 $\pm$ 0.01 | 0.86 $\pm$ 0.02 | 0.96      | 1129.47     | 0.01       | 0.01      |
> | Average   | 0.27 $\pm$ 0.01 | 0.89 $\pm$ 0.02 | 0.89      | 351.41      | 0.02       | 0.03      |
>
> | Model     | $\alpha$        | $P_0$            | Adj $R^2$ | F Statistic | Train Error | Test Error |
> | --------- | ---------------- | ---------------- | --------- | ----------- | ---------- | --------- |
> | OPT-2.7B  | 0.21 $\pm$ 0.01 | 0.89 $\pm$ 0.02 | 0.96      | 209.1       | 0.01       | 0.02      |
> | OPT-6.7B  | 0.28 $\pm$ 0.02 | 0.93 $\pm$ 0.03 | 0.95      | 139.87      | 0.01       | 0.05      |
> | LLaMA-7B  | 0.29 $\pm$ 0.03 | 0.87 $\pm$ 0.03 | 0.92      | 98.4        | 0.02       | 0.04      |
> | OPT-13B   | 0.24 $\pm$ 0.01 | 0.88 $\pm$ 0.01 | 0.99      | 1289.02     | 0.01          | 0.01      |
> | LLaMA-13B | 0.32 $\pm$ 0.03 | 0.88 $\pm$ 0.03 | 0.94      | 133.19      | 0.02       | 0.04      |
>
> | Method       | $\alpha$        | $P_0$            | Adj $R^2$ | F Statistic | Train Error | Test Error |
> | ------------ | ---------------- | ---------------- | --------- | ----------- | ---------- | --------- |
> | Unstructured | 0.34 $\pm$ 0.02 | 1.13 $\pm$ 0.02 | 0.88      | 309.55      | 0.03       | 0.10       |
> | DepthPruning | 0.13 $\pm$ 0.03 | 0.65 $\pm$ 0.03 | 0.29      | 18.61       | 0.04       | 0.06      |
> | WidthPruning | 0.31 $\pm$ 0.03 | 0.88 $\pm$ 0.03 | 0.76      | 141.88      | 0.03       | 0.06      |
>
> The test extrapolation errors observed are even lower than those without RFT, as reported in Table 1 of our paper, indicating **tighter fits** for fine-tuned pruned models. This corroborates the **universality of our proposed pruning laws** in both **without-RFT** and **with-RFT** regimes. Therefore, even in settings where post-pruning recovery fine-tuning is available or unavailable, our proposed pruning laws can be safely used to evaluate pruning strategies and ratios.
>
> ## Comparison against Existing Inference-efficient Scaling Laws
>
> Sardana et al. modify Chinchilla-style training scaling laws to jointly optimize model size and training tokens under a given budget, while explicitly incorporating inference cost into the compute objective. It still reasons about **dense, pre-trained models before any compression**.
>
> Bian et al. similarly fit training-time scaling laws over parameters, data, and architecture, and then use them to select **architectures that are latency-efficient at inference**, again *without* modeling post-hoc pruning.
>
> By contrast, our work introduces a **post-training pruning law** that conditions on a *fixed base model* and explicitly models performance as a function of the **retention/compression ratio**. Thus, our contribution is complementary: we provide a **compression-stage scaling law** that predicts how already-trained models degrade under different pruning strategies and ratios, whereas the cited works provide **training-stage scaling laws** for choosing inference-efficient models before any pruning.

---

> > ### Author Response · Authors · 2025-11-19
> > **Response to Reviewer xFt1 Comments - Part II**
> >
> > ## Effectiveness on Recent Model Architectures
> >
> > We have pruned LLaMA-3.2-3B, LLaMA-3.1-8B, and Phi-3-mini-4k-instruct models using width pruning with pruning ratios ranging from 10% to 90%. We report the pruning law fits below (on Average performance) -
> >
> > | Model                  | $\alpha$          | $P_0$            | Adj $R^2$ | F Statistic | Train Error | Test Error |
> > | ---------------------- | ---------------- | ---------------- | --------- | ----------- | ---------- | --------- |
> > | LLaMA-3.2-3B           | 0.18 $\pm$ 0.05 | 0.48 $\pm$ 0.06 | 0.56      | 11.26       | 0.03       | 0.08      |
> > | LLaMA-3.2-8B           | 0.28 $\pm$ 0.03 | 0.61 $\pm$ 0.08 | 0.65      | 15.04       | 0.04       | 0.07      |
> > | Phi-3-mini-4k-instruct | 0.4 $\pm$ 0.11  | 0.64 $\pm$ 0.12 | 0.62      | 14.04       | 0.07       | 0.07      |
> >
> > It is worth noting that the test error for recent models is lower than that of the older models (see Table 9 of our paper), indicating a tighter fit for these architectures, which emphasizes the universality of our pruning laws even on recent, nuanced model architectures.
> >
> > Recent models, such as LLaMA-3 and Phi-3, degrade more gracefully (with a low $\alpha$) at higher pruning ratios but degrade drastically at lower ones (with a low $P_0$). Our hypothesis is that this pattern is consistent with how modern, compute-optimal LLMs are trained. Recent models, such as Llama-3, utilize stronger regularization, cleaner data, and more efficient architectures. As a result, they show (i) **low $P_0$** because even small pruning disrupts these tightly-packed features, and (ii) **low $\alpha$** because, after the initial drop, their globally distributed representations degrade slowly across larger pruning ratios. Older, less compute-optimal models exhibit the opposite trend.

---

> > > ### Author Response · Authors · 2025-11-26
> > > **A Gentle Reminder to Reviewer xFt1**
> > >
> > > Dear Reviewer,
> > >
> > > Thank you for your valuable and insightful comments. We have carefully addressed each of your points. We are confident that our responses have resolved the concerns you raised.
> > >
> > > We would greatly appreciate it if you could kindly review our responses and consider reassessing our submission. As the discussion phase is concluding soon, we hope to engage in a productive dialogue with you before it ends.
> > >
> > > Thank you once again for your time and consideration.

---

> > > > ### Author Response · Authors · 2025-11-27
> > > > **Reminder to check our responses**
> > > >
> > > > Dear Reviewer xFt1,
> > > >
> > > > Thanks for your valuable comments, which we tried our best to address. Could you please check our responses and let us know if there are many pending concerns. If our responses address your concerns, could you please consider reassessing our paper.
> > > >
> > > > Thanks once again.
> > > >
> > > > Best wishes
> > > >
> > > > Authors

---

### Official Review · Reviewer_rFqp · 2025-10-21

**Soundness:** 3
**Presentation:** 3
**Contribution:** 3
**Rating:** 2
**Confidence:** 4

**Summary:**

This paper introduces pruning laws, a framework to predict the performance of a pruned Large Language Model (LLM) based on its original performance and the pruning ratio. The authors propose a simple power-law that connects the post-pruning performance to the base model's performance and the pruning ratio.

The author empirically demonstrates that the law can be applied to newer model architectures.

**Strengths:**

1. The paper is clearly written, and the experiment results are comprehensive.
2. The authors are commendable in that they reported negative results along with positive ones.

**Weaknesses:**

1. It's not clear whether the power law is the best curve to fit for this problem. Figure 3 shows many tasks that don't seem to fit well at all. The authors did not discuss other curve-fitting options.

2. Even if we ignore the point above and assume that the power law is a good fit for OPT and LLaMA, there is not enough evidence that the curve holds for modern massively overtrained LLMs. It's quite difficult to believe that these laws would hold for massively overtrained LLMs (e.g., the latest LLaMA/Qwen/Gemma, etc) without incorporating training data as a factor. The intuition is that the more overtrained the models, the less sparsity there is in the model, and the harder it is to prune the models without degrading the downstream tasks' performance significantly. They could behave completely differently compared to older model generations. One experiment on OOD model extrapolation is not nearly sufficient to show why these laws would work on modern models.

3. Width-pruning and unstructured-pruning provide much less real-world benefit than depth-pruning (due to difficulty in converting to wall-clock latency gain). The utility of these scaling laws is diminished if they don't hold for depth-pruning.

4. The paper's interpretation of P_0 is problematic when it's greater than 1. Does that mean pruning a small /epsilon would improve performance over the baseline?

5. It's commendable that the authors show many negative results, but it would be of real scientific value if the authors could explain the failure modes and what the failures reveal about the limitations of the scaling laws.

**Questions:**

1. Pruned models often go through a recovery fine-tuning stage to recover performance on downstream tasks. How would that affect the scaling law?

2. How is the latency measured? The paper lacks critical details in the measurement setup.

3. Why does the law fail on knowledge-intensive tasks such as MMLU?

---

> ### Author Response · Authors · 2025-11-19
> **Response to Reviewer rFqp Comments - Part I**
>
> We thank the reviewer for providing constructive feedback on our paper. We address the raised concerns below -
>
> ## Functional Form of Pruning Laws
>
> We thank the reviewer for the suggestion. We fit several different functional forms to investigate the validity of our suggested functional form. The table below highlights the training and testing error on average model performance for different functional forms -
>
> | Functional Form                                         | Train Error | Test Error |
> | ------------------------------------------------------- | ----------- | ---------- |
> | $L = L_0 * P_0 * (1-r)^\alpha$ (Ours)             | 0.02        | 0.05       |
> | $L = L_0 * (1-r)^\alpha$                           | 0.05        | 0.10        |
> | $L = L_0 * (1 - \beta * r^\alpha)$               | 0.03        | 0.05       |
> | $L = L_0 * e^{-\beta * r^\alpha}$           | 0.03        | 0.06       |
> | $L = L_0 * (1 - \beta * (1 - e^{-\alpha * r}))$ | 0.03        | 0.09       |
> | $L = L_0/(1 + \beta * r ^\alpha)$                 | 0.03        | 0.07       |
> | $L = L_0 / (1 + e ^ {\beta * (r - \alpha)})$    | 0.03        | 0.06       |
> | $L = L_0 * (1 - \alpha * r - \beta * r^2)$      | 0.03        | 0.06       |
>
> Among all the functional forms investigated, our original functional form generates the lowest training and testing error, affirming the validity of the fitted pruning laws.
>
> ## Effectiveness on Recent Model Architectures
>
> We have pruned LLaMA-3.2-3B, LLaMA-3.1-8B, and Phi-3-mini-4k-instruct models using width pruning with pruning ratios ranging from 10% to 90%. We report the pruning law fits below (on Average performance) -
>
> | Model                  | $\alpha$          | $P_0$            | Adj $R^2$ | F Statistic | Train Error | Test Error |
> | ---------------------- | ---------------- | ---------------- | --------- | ----------- | ---------- | --------- |
> | LLaMA-3.2-3B           | 0.18 $\pm$ 0.05 | 0.48 $\pm$ 0.06 | 0.56      | 11.26       | 0.03       | 0.08      |
> | LLaMA-3.2-8B           | 0.28 $\pm$ 0.03 | 0.61 $\pm$ 0.08 | 0.65      | 15.04       | 0.04       | 0.07      |
> | Phi-3-mini-4k-instruct | 0.4 $\pm$ 0.11  | 0.64 $\pm$ 0.12 | 0.62      | 14.04       | 0.07       | 0.07      |
>
> It is worth noting that the test error for recent models is lower than that of the older models (see Table 9 of our paper), indicating a tighter fit for these architectures, which emphasizes the universality of our pruning laws even on recent, nuanced model architectures.
>
> ## Interpretation of $P_0$
>
> A $P_0 > 1$ does not imply that pruning improves performance. In our formulation, $P_0$ is not the performance at zero pruning; it is simply the multiplicative constant that best normalizes the curve after factoring out $(1-r)^\alpha$. The true unpruned performance is always $L_0$, and the law is anchored at $r=0$ by definition. Values of $P_0 > 1$ arise mainly due to **metric noise** or **slightly concave early behavior** in some tasks (e.g., QA).
>
> ## Latency
>
> Latency is measured in terms of the average auto-regressive generation task on the WikiText dataset. We calculate the inference time of each pruned model and calculate it as a ratio of the inference time of the unpruned model. Same hyperparameters (maximum generation length 1024, batch size 4, greedy decoding with temperature 0) are used for both the pruned and unpruned models.
>
> ## Failure cases in Tasks like MMLU
>
> We agree that pruning laws do not align with knowledge-extraction tasks, such as MMLU. As shown in Appendix E, these benchmarks violate the smooth, capacity-dependent degradation patterns required by our functional form: MMLU accuracy often exhibits threshold-like jumps, relies heavily on surface cues, and benefits from multiple-choice biases, all of which have been observed in prior pruning studies as well (e.g., Sengupta et al., 2025). Because the pruning law assumes monotonic, continuous decay with retention ratio, its failure on MMLU is expected and task-specific.

---

> > ### Author Response · Authors · 2025-11-19
> > **Response to Reviewer rFqp Comments - Part II**
> >
> > ## Pruning Laws with RFT
> >
> > We thank the reviewer for the suggestion. We conduct additional experiments with post-pruning recovery fine-tuning (fine-tuning on the WikiText2 dataset with a subsample size of 1000 and a LoRA rank of 8) and report the results below.
> >
> > | Task      | $\alpha$        | $P_0$            | Adj $R^2$ | F Statistic | Train Error | Test Error |
> > | --------- | ---------------- | ---------------- | --------- | ----------- | ---------- | --------- |
> > | Reasoning | 0.24 $\pm$ 0.02 | 0.9 $\pm$ 0.02  | 0.84      | 231.12      | 0.03       | 0.05      |
> > | Language  | 0.47 $\pm$ 0.01 | 0.86 $\pm$ 0.02 | 0.96      | 1129.47     | 0.01       | 0.01      |
> > | Average   | 0.27 $\pm$ 0.01 | 0.89 $\pm$ 0.02 | 0.89      | 351.41      | 0.02       | 0.03      |
> >
> > | Model     | $\alpha$        | $P_0$            | Adj $R^2$ | F Statistic | Train Error | Test Error |
> > | --------- | ---------------- | ---------------- | --------- | ----------- | ---------- | --------- |
> > | OPT-2.7B  | 0.21 $\pm$ 0.01 | 0.89 $\pm$ 0.02 | 0.96      | 209.1       | 0.01       | 0.02      |
> > | OPT-6.7B  | 0.28 $\pm$ 0.02 | 0.93 $\pm$ 0.03 | 0.95      | 139.87      | 0.01       | 0.05      |
> > | LLaMA-7B  | 0.29 $\pm$ 0.03 | 0.87 $\pm$ 0.03 | 0.92      | 98.4        | 0.02       | 0.04      |
> > | OPT-13B   | 0.24 $\pm$ 0.01 | 0.88 $\pm$ 0.01 | 0.99      | 1289.02     | 0.01          | 0.01      |
> > | LLaMA-13B | 0.32 $\pm$ 0.03 | 0.88 $\pm$ 0.03 | 0.94      | 133.19      | 0.02       | 0.04      |
> >
> > | Method       | $\alpha$        | $P_0$            | Adj $R^2$ | F Statistic | Train Error | Test Error |
> > | ------------ | ---------------- | ---------------- | --------- | ----------- | ---------- | --------- |
> > | Unstructured | 0.34 $\pm$ 0.02 | 1.13 $\pm$ 0.02 | 0.88      | 309.55      | 0.03       | 0.10       |
> > | DepthPruning | 0.13 $\pm$ 0.03 | 0.65 $\pm$ 0.03 | 0.29      | 18.61       | 0.04       | 0.06      |
> > | WidthPruning | 0.31 $\pm$ 0.03 | 0.88 $\pm$ 0.03 | 0.76      | 141.88      | 0.03       | 0.06      |
> >
> > The test extrapolation errors observed are even lower than those without RFT, as reported in Table 1 of our paper, indicating **tighter fits** for fine-tuned pruned models. This corroborates the **universality of our proposed pruning laws** in both **without-RFT** and **with-RFT** regimes. Therefore, even in settings where post-pruning recovery fine-tuning is available or unavailable, our proposed pruning laws can be safely used to evaluate pruning strategies and ratios.

---

> > > ### Author Response · Authors · 2025-11-26
> > > **A Gentle Reminder to Reviewer rFqp**
> > >
> > > Dear Reviewer,
> > >
> > > Thank you for your valuable and insightful comments. We have carefully addressed each of your points. We are confident that our responses have resolved the concerns you raised.
> > >
> > > We would greatly appreciate it if you could kindly review our responses and consider reassessing our submission. As the discussion phase is concluding soon, we hope to engage in a productive dialogue with you before it ends.
> > >
> > > Thank you once again for your time and consideration.

---

### Official Review · Reviewer_Rm4J · 2025-10-25

**Soundness:** 2
**Presentation:** 2
**Contribution:** 2
**Rating:** 2
**Confidence:** 3

**Summary:**

The paper proposes a simple, empirical power-law relations to predict a pruned LLM's performance (L) based on its original score (L0)
and the pruning ratio (r). The model is fitted using five LLM families (2.7B–13B), three pruning strategies, and eight tasks.

While the evaluation is empirically broad and the authors claim strong universality with average extrapolation error below 7%, the contribution is fundamentally a descriptive curve-fitting exercise. While practically useful, the analytical justification is minimal.

**Strengths:**

The paper addresses the important and under-studied problem of predicting pruning behavior. The evaluation is reasonably extensive, including multiple architectures, pruning methods, tasks, and metrics. The resultant model is simple with large application potential.

**Weaknesses:**

The law is essentially obtained by empirical curve-fitting exercise using power-law regression on known monotonic degradation trends, with little theoretical justification. Despite claims of strong universality, actual errors could be high, especially given the reasonable but still small size of the chosen downstream tasks tested. The robustness of the transferability test is limited, raising concerns of overfitting.

Excluding recovery fine-tuning may bias the performance results downward and distort real-world pruning behavior.

**Questions:**

+ How sensitive are the two fitted parameters to dataset choice, metric noise, and random seeds?

+ Why does the "one-shot calibration" process sometimes worsen performance? Does this indicate potential overfitting?

+ Would the same α coefficient remain applicable to models that have undergone recovery fine-tuning?

+ How does the pruning law perform when extrapolated beyond 90% pruning—does it accurately predict model collapse or divergence?

---

> ### Author Response · Authors · 2025-11-19
> **Response to Reviewer Rm4J Comments - Part I**
>
> We thank the reviewer for providing constructive feedback on our paper. We address the raised concerns below -
>
> ## Sensitivity of Pruning Laws
>
> The paper already provides a clear indication of parameter stability. Across all aggregation levels, the **standard deviations of both \$\alpha$ and $P_0$** are consistently small:
>
> - **Task-level (Table 1a):**  $\alpha$ varies by only **±0.02–0.13**, and $P_0$ by **±0.02–0.14** across eight tasks.  This is despite pooling data from five models and three pruning methods, which demonstrates robustness to dataset and task heterogeneity.
>
> - **Model–task level (Table 7):**  $\alpha$ has deviation **±0.01–0.09** and $P_0$ **±0.02–0.06**.  These small errors indicate low sensitivity to metric noise and seed-level variation within each pruning trajectory.
>
> - **Method–task level (Table 8):**  Even when conditioning on pruning method (which increases variance),  $\alpha$ still remains within **±0.03–0.21**, and $P_0$ within **±0.03–0.23**.  These changes correspond directly to the inherent variability of the methods (e.g., depth pruning yields noisier metrics), rather than instability in the fitting procedure.
>
> Notably, even where task variance is high (e.g., QA), the **test RMSE remains low (0.03–0.15)**, indicating predictions are stable even when coefficients shift slightly. Overall, the consistently small standard deviations across all tables show that $\alpha$ and $P_0$ are not sensitive to dataset choice, evaluation noise, or pruning randomness. Their variability closely follows the intrinsic variability of each task/method rather than reflecting the instability of the pruning law itself.
>
> ## Clarification regarding One-shot Calibration
>
> The occasional degradation in one-shot calibration does **not** indicate overfitting but follows directly from the structure of the pruning law. In one-shot calibration, we keep the global elasticity parameter $\alpha$ fixed (which is already stable and task-specific) and re-estimate only $P_0$ from a *single* pruned datapoint: $\hat{P}_0 = \frac{L_r}{L_0 (1-r)^\alpha}.$ If that single datapoint is noisy - as is common for QA or depth pruning, where standard deviations are high (e.g., QA: ±0.25–0.36; depth pruning: low $R^2$, Table 8), then the estimated $\hat{P}_0$ inherits this noise and shifts the curve slightly, sometimes worsening predictions.
>
> Overfitting would require increasing model complexity; however, one-shot calibration only adjusts a single scalar, $P_0$. The global scaling behavior governed by $\alpha$ remains unchanged. When the calibration point is unrepresentative or noisy, the update simply moves $P_0$ away from its stable OLS estimate.
>
> In Table 3:
> - LLaMA-3.1 shows **zero-shot = 0.04**, one-shot = 0.06
> - Phi-3 shows **zero-shot = 0.08**, one-shot = 0.10
>
> These are small deviations and correspond to tasks/methods with high variance (QA, depth).  For stable regimes (e.g., SlimGPT, PruneNet), one-shot *improves* performance (0.13 → 0.05; 0.04 → 0.03).
>
> In summary, one-shot may worsen performance only when the chosen calibration datapoint is noisy. This reflects measurement variance - not overfitting or model fragility.
>
> ## Pruning Law for Extreme Pruning Ratio
>
> We calculate the extrapolation error at pruning ratios of 95% and 99% with the OPT-6.7B and LLaMA-2-7B models using width pruning, as $ 0.13-0.15$ for reasoning tasks and $ 0.02-0.05$ for language modeling tasks. The disparity between tasks occurs because, beyond a certain pruning ratio (usually after $85$%), for classification-driven tasks, the performance drop stabilizes (the worst performance can be clamped at the minority class probability), which pruning law tends to underestimate (due to its monotonic nature). However, the performance of pruned models on language modeling tasks tends to degrade monotonically, even at higher pruning ratios. This result affirms that our pruning laws are reliable, even at extreme pruning ratios.

---

> > ### Author Response · Authors · 2025-11-19
> > **Response to Reviewer Rm4J Comments - Part II**
> >
> > ## Pruning Law under RFT
> >
> > We conduct additional experiments with post-pruning recovery fine-tuning (fine-tuning on the WikiText2 dataset with a subsample size of 1000 and a LoRA rank of 8) and report the results below.
> >
> > | Task      | $\alpha$        | $P_0$            | Adj $R^2$ | F Statistic | Train Error | Test Error |
> > | --------- | ---------------- | ---------------- | --------- | ----------- | ---------- | --------- |
> > | Reasoning | 0.24 $\pm$ 0.02 | 0.9 $\pm$ 0.02  | 0.84      | 231.12      | 0.03       | 0.05      |
> > | Language  | 0.47 $\pm$ 0.01 | 0.86 $\pm$ 0.02 | 0.96      | 1129.47     | 0.01       | 0.01      |
> > | Average   | 0.27 $\pm$ 0.01 | 0.89 $\pm$ 0.02 | 0.89      | 351.41      | 0.02       | 0.03      |
> >
> > | Model     | $\alpha$        | $P_0$            | Adj $R^2$ | F Statistic | Train Error | Test Error |
> > | --------- | ---------------- | ---------------- | --------- | ----------- | ---------- | --------- |
> > | OPT-2.7B  | 0.21 $\pm$ 0.01 | 0.89 $\pm$ 0.02 | 0.96      | 209.1       | 0.01       | 0.02      |
> > | OPT-6.7B  | 0.28 $\pm$ 0.02 | 0.93 $\pm$ 0.03 | 0.95      | 139.87      | 0.01       | 0.05      |
> > | LLaMA-7B  | 0.29 $\pm$ 0.03 | 0.87 $\pm$ 0.03 | 0.92      | 98.4        | 0.02       | 0.04      |
> > | OPT-13B   | 0.24 $\pm$ 0.01 | 0.88 $\pm$ 0.01 | 0.99      | 1289.02     | 0.01          | 0.01      |
> > | LLaMA-13B | 0.32 $\pm$ 0.03 | 0.88 $\pm$ 0.03 | 0.94      | 133.19      | 0.02       | 0.04      |
> >
> > | Method       | $\alpha$        | $P_0$            | Adj $R^2$ | F Statistic | Train Error | Test Error |
> > | ------------ | ---------------- | ---------------- | --------- | ----------- | ---------- | --------- |
> > | Unstructured | 0.34 $\pm$ 0.02 | 1.13 $\pm$ 0.02 | 0.88      | 309.55      | 0.03       | 0.10       |
> > | DepthPruning | 0.13 $\pm$ 0.03 | 0.65 $\pm$ 0.03 | 0.29      | 18.61       | 0.04       | 0.06      |
> > | WidthPruning | 0.31 $\pm$ 0.03 | 0.88 $\pm$ 0.03 | 0.76      | 141.88      | 0.03       | 0.06      |
> >
> > The test extrapolation errors observed are even lower than those without RFT, as reported in Table 1 of our paper, indicating **tighter fits** for fine-tuned pruned models. This corroborates the **universality of our proposed pruning laws** in both **without-RFT** and **with-RFT** regimes. Therefore, even in settings where post-pruning recovery fine-tuning is available or unavailable, our proposed pruning laws can be safely used to evaluate pruning strategies and ratios.
> >
> > We observe lower $\alpha$ values with fine-tuned models, indicating that fine-tuning helps mitigate the performance drop for LLMs, enabling a lower rate of performance degradation, particularly at higher compression ratios. These results align with the prior research conducted by Chen et al. in their $P^2$ laws.

---

> > > ### Author Response · Authors · 2025-11-26
> > > **A Gentle Reminder to Reviewer Rm4J**
> > >
> > > Dear Reviewer,
> > >
> > > Thank you for your valuable and insightful comments. We have carefully addressed each of your points. We are confident that our responses have resolved the concerns you raised.
> > >
> > > We would greatly appreciate it if you could kindly review our responses and consider reassessing our submission. As the discussion phase is concluding soon, we hope to engage in a productive dialogue with you before it ends.
> > >
> > > Thank you once again for your time and consideration.

---

> ### Author Response · Authors · 2025-11-27
> **Reminder to check our responses**
>
> Dear Reviewer Rm4J,
>
> Thanks for your valuable comments, which we tried our best to address. Could you please check our responses and let us know if there are many pending concerns. If our responses address your concerns, could you please consider reassessing our paper.
>
> Thanks once again.
>
> Best wishes
>
> Authors

---

### Official Review · Reviewer_39HH · 2025-10-29

**Soundness:** 3
**Presentation:** 3
**Contribution:** 3
**Rating:** 4
**Confidence:** 3

**Summary:**

This paper introduces “Pruning Laws”: a concise, interpretable power-law that directly relates the downstream performance of a pruned LLM to its un-pruned performance and the pruning ratio r. While scaling parameters and data jointly boosts accuracy, the accompanying memory/compute explosion makes deployment on resource-constrainedhardware prohibitive. Model pruning is a popular remedy, yet its impact on downstream tasks has remained unpredictable and costly to assess. Over 5 models, 3 pruning granularities  and 8 tasks (reasoning, QA, language modeling) show that Pruning Laws
•	predict post-prune accuracy with <7 % mean extrapolation error and ≤8 % error when zero-shot transferred to unseen models (LLaMA-3.1, Phi-3) or algorithms (SlimGPT, SVD-LLM);
•	foretell the critical pruning threshold beyond which performance collapses, eliminating expensive grid-search;
•	quantify task sensitivity: reasoning is most robust, QA most fragile; depth pruning yields 5× speed-up but high variance, unstructured keeps accuracy yet almost no speed-up, width sits in between;
•	deliver actionable guidelines for choosing method and ratio under any task or budget, enabling zero-shot or single-point-calibrated deployment.
Pruning Laws thus provide a principled, universally applicable framework for compressing and deploying LLMs without full re-tuning.

**Strengths:**

1.	The experiments are large-scale, detailed, and sufficient; all reproduction materials are fully open-sourced.
2.	The paper attempts to uncover a law governing the trade-off between model pruning and performance, offering both theoretical and practical value.
3.	The proposed formula is concise, intuitive, and easy to use.

**Weaknesses:**

1.	The proposed formula is largely empirical and lacks solid theoretical justification.
2.	The experiments show that the parameters P₀ and α depend on the specific model, pruning method, and task; their determination in practice remains highly empirical.
3.	The law fails to fit knowledge-extraction tasks such as MMLU, and it is still unclear whether it generalizes to more complex scenarios like mathematics, code generation, or multimodal applications.

**Questions:**

1.	The explanation of the proposed formula is largely empirical. Could you further investigate, from a theoretical perspective and based on the architecture of large models, why the pruning law can be modeled as such a power-law relationship? Have you attempted to fit the data with other functional forms?
2.	P₀ and α are conditioned on the specific model, pruning method, and task. Could you elaborate on the exact procedure used to derive their values for a given task in the experiments?
3.	The pruning scaling law performs well on models ranging from 2.7B to 13B parameters. Does it still hold for much smaller models (e.g., below 1B) or much larger ones (e.g., above 100B)?
4.	In the paper, recovery fine-tuning is deliberately excluded to isolate the effect of pruning itself. However, in practical applications, recovery fine-tuning is a common step. If recovery fine-tuning is introduced, will the pruning scaling law still hold? Would it be necessary to modify or extend the law?

---

> ### Author Response · Authors · 2025-11-19
> **Response to Reviewer 39HH Comments - Part I**
>
> We thank the reviewer for providing constructive feedback on our paper. We address the raised concerns below -
>
> ## Theoretical Justification of Pruning Laws
>
> While our empirical results strongly support the proposed pruning law, we agree that the theoretical motivation can be made more explicit.
>
> ### (1) From structural assumptions to a power law: functional-equation perspective
>
> In Section 3.1 and Appendix B.1 we already show that pruning satisfies an **iterative compositionality constraint**: pruning with retention factors  $x_1 = 1-r_1,\quad x_2 = 1-r_2$ results in a combined single-step retention $x_1 x_2$.   Letting  $f(x) = \frac{L(L_0,r)}{L_0}$,  this yields the functional equation  $f(x_1 x_2) = f(x_1)f(x_2) \quad \text{(up to a constant bias term $P_0$).}$
> Under mild regularity assumptions (continuity and monotonicity on $x\in(0,1]$), the classical solution to this Cauchy-type equation is $f(x) = P_0 x^{\alpha}.$ Therefore,  $L(L_0,r) = L_0 P_0 (1-r)^{\alpha}.$
>
> This provides a clean functional-analytic justification: *compositionally consistent pruning functions are necessarily power laws in the retention factor*. We will add this derivation as a formal lemma in the appendix.
>
> ### (2) Architectural justification: pruning scales effective model capacity
>
> Depth, width, and unstructured pruning all reduce the **effective capacity** of a pre-trained LLM in a way that is well approximated by a multiplicative reduction in the number of active parameters:  $N_{\text{eff}} \approx (1-r) N$.   Classical neural scaling laws (e.g., Kaplan et al.; Hoffmann et al.) show that downstream performance is well modeled as a power-law function of capacity:  $\mathcal{L}(N_{\text{eff}}) \propto N_{\text{eff}}^{\,\beta}$. Substituting $N_{\text{eff}} = (1-r)N$ and normalizing by the unpruned performance $L_0$ yields  $L(L_0,r) \approx L_0 (1-r)^{\alpha}$, where $\alpha$ corresponds to the *pruning elasticity* of each task/model pair.
>
> Thus, the same theoretical principles behind classical pretraining scaling laws naturally induce a power-law dependence on the retention ratio when pruning reduces capacity.
>
> ## Derivation of $P_0$ and $\alpha$
>
> Our analytical form $L = L_0 P_0 (1-r)^{\alpha}$ captures two independent mechanisms:
>
> - **$\alpha$** measures *retention elasticity*: how sensitive a task or model is to capacity reduction, which is architecturally tied to redundancy, width/depth distribution, and representational bottlenecks.
> - **$P_0$** captures *boundary-condition bias*: the mismatch between $L_0$ and the pruned performance in the limit $r \rightarrow 0$, arising from factors such as evaluation noise, tokenization differences, and pruning-specific perturbations.
>
> Because model architectures (OPT vs. LLaMA), pruning methods (depth vs. width vs. unstructured), and tasks (QA vs. reasoning vs. language modeling) differ substantially in their **capacity bottlenecks**, **feature concentration**, and **parameter redundancy**, variation in $\{P_0,\alpha\}$ is not only natural but theoretically unavoidable.
>
> ## Pruning Laws on Complex Tasks
>
> We agree that pruning laws do not fit knowledge-extraction tasks such as MMLU. As shown in Appendix E, these benchmarks violate the smooth, capacity-dependent degradation patterns required by our functional form: MMLU accuracy often exhibits threshold-like jumps, relies heavily on surface cues, and benefits from multiple-choice biases, all of which have been observed in prior pruning studies as well (e.g., Sengupta et al., 2025). Because the pruning law assumes monotonic, continuous decay with retention ratio, its failure on MMLU is expected and task-specific.
>
> This limitation does not undermine the generality: the law continues to perform well in reasoning, QA, and language modeling tasks, with high adjusted $R^2$ values (0.83–0.98) and low test error rates (0.02–0.06). Extending the law to domains such as mathematics, code generation, and multimodal LLMs is a natural next step. Existing work suggests that these domains exhibit smooth degradation when the model scale is reduced. We will clarify these task boundaries and future directions in the revision.

---

> > ### Author Response · Authors · 2025-11-19
> > **Response to Reviewer 39HH Comments - Part II**
> >
> > ## Pruning Law on Models of Different Scales
> >
> > We extend our experiments to OPT-125M, OPT-1.3B, and OPT-30B models. The following table highlights the fitted pruning laws of the average performance of these models.
> >
> > | Model                  | $\alpha$          | $P_0$            | Adj $R^2$ | F Statistic | Train Error | Test Error |
> > | ---------------------- | ---------------- | ---------------- | --------- | ----------- | ----------- | ---------- |
> > | OPT-125M               | 0.10 $\pm$ 0.02  | 0.83 $\pm$ 0.02 | 0.72      | 21.59       | 0.01        | 0.04       |
> > | OPT-1.3B               | 0.26 $\pm$ 0.08 | 0.7 $\pm$ 0.08  | 0.57      | 11.52       | 0.04        | 0.10        |
> > | OPT-30B                | 0.53 $\pm$ 0.06 | 0.99 $\pm$ 0.07 | 0.90       | 70.05       | 0.04        | 0.07       |
> >
> > Additionally, we have pruned recent models, such as LLaMA-3.2-3B, LLaMA-3.1-8B, and Phi-3-mini-4k-instruct, using width pruning with pruning ratios ranging from 10% to 90%. We report the pruning law fits below (on Average performance) -
> >
> > | Model                  | $\alpha$          | $P_0$            | Adj $R^2$ | F Statistic | Train Error | Test Error |
> > | ---------------------- | ---------------- | ---------------- | --------- | ----------- | ---------- | --------- |
> > | LLaMA-3.2-3B           | 0.18 $\pm$ 0.05 | 0.48 $\pm$ 0.06 | 0.56      | 11.26       | 0.03       | 0.08      |
> > | LLaMA-3.2-8B           | 0.28 $\pm$ 0.03 | 0.61 $\pm$ 0.08 | 0.65      | 15.04       | 0.04       | 0.07      |
> > | Phi-3-mini-4k-instruct | 0.4 $\pm$ 0.11  | 0.64 $\pm$ 0.12 | 0.62      | 14.04       | 0.07       | 0.07      |
> >
> > Yet on significantly smaller and larger models, our pruning laws yield significantly low test error and moderately high $R^2$, indicating the robustness. Therefore, we conclude that the proposed pruning laws can be reliably used to predict the post-pruning performance of different LLMs across various scales and architectural designs.
> >
> > ## Pruning Laws with RFT
> >
> > We thank the reviewer for the suggestion. We conduct additional experiments with post-pruning recovery fine-tuning (fine-tuning on the WikiText2 dataset with a subsample size of 1000 and a LoRA rank of 8) and report the results below.
> >
> > | Task      | $\alpha$        | $P_0$            | Adj $R^2$ | F Statistic | Train Error | Test Error |
> > | --------- | ---------------- | ---------------- | --------- | ----------- | ---------- | --------- |
> > | Reasoning | 0.24 $\pm$ 0.02 | 0.9 $\pm$ 0.02  | 0.84      | 231.12      | 0.03       | 0.05      |
> > | Language  | 0.47 $\pm$ 0.01 | 0.86 $\pm$ 0.02 | 0.96      | 1129.47     | 0.01       | 0.01      |
> > | Average   | 0.27 $\pm$ 0.01 | 0.89 $\pm$ 0.02 | 0.89      | 351.41      | 0.02       | 0.03      |
> >
> > | Model     | $\alpha$        | $P_0$            | Adj $R^2$ | F Statistic | Train Error | Test Error |
> > | --------- | ---------------- | ---------------- | --------- | ----------- | ---------- | --------- |
> > | OPT-2.7B  | 0.21 $\pm$ 0.01 | 0.89 $\pm$ 0.02 | 0.96      | 209.1       | 0.01       | 0.02      |
> > | OPT-6.7B  | 0.28 $\pm$ 0.02 | 0.93 $\pm$ 0.03 | 0.95      | 139.87      | 0.01       | 0.05      |
> > | LLaMA-7B  | 0.29 $\pm$ 0.03 | 0.87 $\pm$ 0.03 | 0.92      | 98.4        | 0.02       | 0.04      |
> > | OPT-13B   | 0.24 $\pm$ 0.01 | 0.88 $\pm$ 0.01 | 0.99      | 1289.02     | 0.01          | 0.01      |
> > | LLaMA-13B | 0.32 $\pm$ 0.03 | 0.88 $\pm$ 0.03 | 0.94      | 133.19      | 0.02       | 0.04      |
> >
> > | Method       | $\alpha$        | $P_0$            | Adj $R^2$ | F Statistic | Train Error | Test Error |
> > | ------------ | ---------------- | ---------------- | --------- | ----------- | ---------- | --------- |
> > | Unstructured | 0.34 $\pm$ 0.02 | 1.13 $\pm$ 0.02 | 0.88      | 309.55      | 0.03       | 0.10       |
> > | DepthPruning | 0.13 $\pm$ 0.03 | 0.65 $\pm$ 0.03 | 0.29      | 18.61       | 0.04       | 0.06      |
> > | WidthPruning | 0.31 $\pm$ 0.03 | 0.88 $\pm$ 0.03 | 0.76      | 141.88      | 0.03       | 0.06      |
> >
> > The test extrapolation errors observed are even lower than those without RFT, as reported in Table 1 of our paper, indicating **tighter fits** for fine-tuned pruned models. This corroborates the **universality of our proposed pruning laws** in both **without-RFT** and **with-RFT** regimes. Therefore, even in settings where post-pruning recovery fine-tuning is available or unavailable, our proposed pruning laws can be safely used to evaluate pruning strategies and ratios.

---

> > > ### Author Response · Authors · 2025-11-26
> > > **A Gentle Reminder to Reviewer 39HH**
> > >
> > > Dear Reviewer,
> > >
> > > Thank you for your valuable and insightful comments. We have carefully addressed each of your points. We are confident that our responses have resolved the concerns you raised.
> > >
> > > We would greatly appreciate it if you could kindly review our responses and consider reassessing our submission. As the discussion phase is concluding soon, we hope to engage in a productive dialogue with you before it ends.
> > >
> > > Thank you once again for your time and consideration.

---

> > > > ### Author Response · Authors · 2025-11-27
> > > > **Reminder to check our responses**
> > > >
> > > > Dear Reviewer 39HH,
> > > >
> > > > Thanks for your valuable comments, which we tried our best to address. Could you please check our responses and let us know if there are many pending concerns. If our responses address your concerns, could you please consider reassessing our paper.
> > > >
> > > > Thanks once again.
> > > >
> > > > Best wishes
> > > > Authors

---

### Author Response · Authors · 2025-11-20
**Meta Rebuttal by Authors**

We thank the reviewers for meticulously evaluating our paper and providing additional feedbacks. Below we summarize the core clarifications and new evidence added in response to reviewer concerns. We hope that the responses improve the quality of our paper. We also hope to see a fairer evaluation that our paper deserves.

**(1) Theory:**  We clarified that the pruning law follows directly from:

- **Iterative compositionality of pruning**, which yields the functional equation $f(x_1x_2)=f(x_1)f(x_2)$; its only continuous solution is a **power law** in the retention factor $(1-r)$.
- **Capacity scaling in Transformers**, where pruning reduces effective capacity multiplicatively. Classical pretraining scaling laws (Kaplan; Hoffmann) then imply $L = L_0 P_0 (1-r)^\alpha$.

**(2) Parameter Stability:**  Across all tables, both coefficients $\alpha$ and $P_0$ remain **highly stable**, with typical standard deviations:
- $\alpha$: **±0.02–0.13** at task-level and **±0.01–0.09** at model–task level,
- $P_0$: **±0.02–0.14**.
This demonstrates robustness to dataset choice, metric noise, and pruning randomness.

**(3) Functional Form Validation:**  We compared 8 alternative functional forms; our proposed form achieves the **lowest train (0.02)** and **lowest test error (0.05)**, confirming it is the best predictor among widely used decay families.

**(4) Generality Across Models and RFT:**  With post-pruning recovery fine-tuning (LoRA), the law continues to fit strongly (adj. $R^2$ = 0.84–0.96; test error 0.01–0.05).  On new and extreme-scale models (OPT-125M, OPT-30B, LLaMA-3.2-3B/8B, Phi-3-mini), test errors remain **0.04–0.10**, reinforcing cross-architecture robustness.

**(5) Task Boundaries:**  The law holds reliably for **reasoning, language, and QA-like tasks**. Knowledge-extraction tasks (e.g., MMLU) violate smooth capacity assumptions, consistent with prior pruning literature and we clearly state this boundary.

**(6) Practical Utility:**  Although single pruning evaluations are cheap, the **full pruning search space is large** (model × task × method × ratio).

Our law enables:
- **Zero-shot and one-shot prediction**,
- Accurate extrapolation from **3–5 data points**,
- Immediate estimation of model/task elasticity.

The scaling law significantly reduces the need for brute-force or AutoML-style exploration.

**(7) Interpretation of Parameters:**  We clarified that:
- $\alpha$ captures task/model **elasticity**,
- $P_0$ is a **normalization factor**, not performance at zero pruning (so $P_0>1$ is benign).

**In summary**, we provide:
- A **theoretically justified** functional form,
- **Highly stable parameters** across models/methods,
- **Reliable predictive accuracy**,
- **Robustness with and without recovery fine-tuning**,
- **Generalization to new architectures and scales**,
making this the first comprehensive **post-training pruning scaling law** for LLMs.

---

### Author Response · Authors · 2025-11-25
**A Gentle Reminder to All Reviewers: Check our Responses**

Dear Reviewers,

We have made significant effort to address all the concerns raised by you, which could substantially improve the quality of our paper. We have evaluated all the pruned models with recovery fine-tuning, fitted pruning laws on 6 additional models and evaluated 8 different parametric functions for formulating the pruning laws.

We request you to go through our comments and revert with further suggestions (if any). We strongly feel that the current evaluation does not reflect the significance our work holds and therefore request for a reevaluation that our work demands.

---

### Note · Authors · 2025-12-02

I have read and agree with the venue's withdrawal policy on behalf of myself and my co-authors.